_Article_

EMBO
Molecular Medicine

# RIG-I is an intracellular checkpoint that limits CD8+ T-cell antitumour immunity

Xiaobing Duan [ID][1,2,12][✉], Jiali Hu [ID][1,12], Yuncong Zhang[1,12], Xiaoguang Zhao[1], Mingqi Yang[1], Taoping Sun [ID][3],
Siya Liu[4], Xin Chen[2], Juan Feng[2], Wenting Li[1], Ze Yang[1], Yitian Zhang [ID][1], Xiaowen Lin[1], Dingjie Liu[1],
Ya Meng[1], Guang Yang[1], Qiuping Lin[3], Guihai Zhang [ID][5], Haihong Lei[6], Zhengsheng Yi[6], Yanyan Liu[1],
Xiaobing Liang[7], Yujuan Wu[8], Wenqing Diao[8], Zesong Li [ID][9], Haihai Liang [ID][1,10], Meixiao Zhan[1,11],
Hong-Wei Sun [ID][1][✉], Xian-Yang Li [ID][1][✉] & Ligong Lu [ID][1,11][✉]

## Abstract

**Retinoic acid-inducible gene I (RIG-I) is a pattern recognition receptor involved in innate immunity, but its role in adaptive immunity, specifically in the context of CD8+ T-cell antitumour immunity, remains unclear. Here, we demonstrate that RIG-I is upregulated in tumour-infiltrating CD8+ T cells, where it functions as an intracellular checkpoint to negatively regulate CD8+ T-cell function and limit antitumour immunity. Mechanistically, the upregulation of RIG-I in CD8+ T cells is induced by activated T cells, and directly inhibits the AKT/glycolysis signalling pathway. In addition, knocking out RIG-I enhances the efficacy of adoptively transferred T cells against solid tumours, and inhibiting RIG-I enhances the response to PD-1 blockade. Overall, our study identifies RIG-I as an intracellular checkpoint and a potential target for alleviating inhibitory constraints on T cells in cancer immunotherapy, either alone or in combination with an immune checkpoint inhibitor.**

**Keywords** RIG-I; CD8+ T cells; Immune Checkpoint; AKT/Glycolysis Signalling Pathway; Immunotherapy
**Subject Categories** Cancer; Immunology

## Introduction

Immune checkpoints are vital for maintaining immune homoeostasis and effective anti-tumour responses. During pathogen invasion, immune cells are activated through antigenic and costimulatory signals presented by antigen-presenting cells (APCs). Simultaneously, a series of coinhibitory receptors is upregulated to produce coinhibitory signals that terminate immune responses and prevent excessive damage to normal tissues. Tumour cells exploit this mechanism by continuously overamplifying immune checkpoint signals, leading to tumour immune escape, which provides a potential target for immune checkpoint inhibitor therapy. Several immune checkpoint inhibitors, such as inhibitors of PD-1, CTLA-4, TIGIT, TIM-3 and LAG-3, are undergoing clinical trials or have been approved for clinical use (Curigliano et al, 2021; Freed-Pastor et al, 2021; Kwon et al, 2021; Shitara et al, 2022; Thudium et al, 2022). However, the low response rate of therapeutic monoclonal antibodies, such as PD-1 antibodies, in the treatment of malignant solid tumours limits their clinical application (Yin et al, 2021). Therefore, exploring new and broader immune checkpoints targeting CD8+ T cells to increase the therapeutic efficacy and improve the response rate will provide a novel therapeutic strategy for the clinical treatment of refractory malignant solid tumours. Additionally, while immune checkpoint receptors are considered cell-surface molecules, emerging evidence has shown that intrinsic proteins such as PTP1B and SOCS1 are also critical regulators of T-cell anti-tumour function and have been identified as intracellular checkpoints (Sutra Del Galy et al, 2021; Wiede et al, 2022). To date, our knowledge of intracellular checkpoints and their translational potential is still limited.

[1]Guangdong Provincial Key Laboratory of Tumour Interventional Diagnosis and Treatment, Zhuhai People's Hospital (Zhuhai Clinical Medical College of Jinan University), Zhuhai 519000, China. [2]Gene Editing Technology Center of Guangdong Province, School of Medicine, Foshan University, Foshan 528225, China. [3]Zhuhai Precision Medical Center, Zhuhai People's Hospital (Zhuhai Clinical Medical College of Jinan University), Zhuhai 519000, China. [4]The Third People's Hospital of Zhuhai, Zhuhai 519000, China. [5]Department of Oncology, Zhuhai People's Hospital (Zhuhai Clinical Medical College of Jinan University), Zhuhai 519000, China. [6]Department of Radiation Oncology, Zhuhai People's Hospital (Zhuhai Clinical Medical College of Jinan University), Zhuhai 519000, China. [7]Guangdong Huixin Life Science Co., Ltd., Zhuhai 519000, China. [8]Zhuhai Central Blood Station, Zhuhai 519000, China. [9]Guangdong Provincial Key Laboratory of Systems Biology and Synthetic Biology for Urogenital Tumours, Shenzhen Key Laboratory of Genitourinary Tumour, Department of Urology, The First Affiliated Hospital of Shenzhen University, Shenzhen Second People's Hospital (Shenzhen Institute of Translational Medicine), Shenzhen, China. [10]State Key Laboratory of Frigid Zone Cardiovascular Diseases (SKLFZCD), Department of Pharmacology (State-Province Key Laboratories of Biomedicine-Pharmaceutics of China, Key Laboratory of Cardiovascular Research, Ministry of Education), College of Pharmacy, Harbin Medical University, Harbin 150081, China. [11]Guangzhou First Pepople's Hospital, the Second Affiliated Hospital, School of Medicine, South China University of Technology, Guangzhou 510006, China. [12]These authors contributed equally: Xiaobing Duan, Jiali Hu, Yuncong Zhang. ✉E-mail: duanxb3@mail3.sysu.edu.cn; shongw@mail3.sysu.edu.cn; lixianyang@ext.jnu.edu.cn; luligong1969@jnu.edu.cn

RIG-I, also known as retinoic acid-inducible gene I protein, is a classical pathogen pattern recognition receptor that is essential for innate immunity. It recognises three RNA structures (5′ triphosphate double-stranded RNA, 5′ capless diphosphate RNA and 5′ 2′-O demethylated RNA) through the carboxyl-terminal domain (CTD), leading to structural transformation and activation of the downstream MAVS/TBK1/IRF3/IRF7/NF-κB antiviral signalling pathway (Onomoto et al, 2021; Song et al, 2022). RIG-I can also bind to specific host RNAs, such as the host's noncoding RNA, after viral infection, leading to RIG-I activation (Chiang et al, 2018). Due to the important role of RIG-I in innate immunity, numerous studies have reported that RIG-I agonists, either alone or in combination with other drugs, have potent anti-tumour effects on a range of tumour models. Activation of the innate immune function of RIG-I in tumours enhances immune checkpoint inhibitor-mediated cancer immunotherapy (Heidegger et al, 2019a; Heidegger et al, 2019b), promotes lymphocyte infiltration into the tumour, and reduces tumour growth and metastasis (Ellermeier et al, 2013). Furthermore, as a multifunctional molecule, RIG-I is involved not only in the antiviral response but also in other regulatory processes in the body, such as cellular apoptosis, immune homoeostasis, and anti-tumour and protumour processes (Heidegger et al, 2019a; Jiang et al, 2023; Wang et al, 2022; Yang et al, 2017). For example, RIG-I can activate the STAT1 signalling pathway to inhibit the proliferation of tumour cells (Jiang et al, 2011); however, the effects of RIG-I on different cell types and cellular components are diverse and even contradictory.

T-cell exhaustion is a phenotype or state of T cells under prolonged antigen stimulation, leading to the gradual loss of cytokines, high expression of inhibitory receptors, metabolic changes, decreased proliferation potential, and reduced survival (Belk et al, 2022; Franco et al, 2020). Exhausted T cells are the main target cell population for immune checkpoint inhibitor therapy in cancer patients. An increasing number of related studies have confirmed that changes in the transcriptome of exhausted T cells are accompanied by overall changes in signalling pathways and epigenetics, leading to the suppression of effective immune responses and the ineffectiveness of immune checkpoint therapy (de Miguel and Calvo 2020; Sun et al, 2023). Therefore, exploring the mechanism of RIG-I as a new intracellular immune checkpoint to resist CD8$^+$ T cells and develop immunotherapy targets against RIG-I for exhausted T cells is necessary. In addition to its classical role in innate immunity, RIG-I also plays a role in regulating the anti-tumour response and immune homoeostasis (Jiang et al, 2023; Yang et al, 2017). When the CARD domain of RIG-I binds to the SH1 domain of Src, the PxxP motif of RIG-I, competes with the PxxP motif of AKT, for binding to the SH3 domain of Src, which prevents Src from activating AKT, thus inhibiting the stemness of leukaemia cells (Li et al, 2014). Additionally, the PI3K/AKT/glycolysis signalling pathway has been shown to be crucial for CD8$^+$ T cells to exert anti-tumour effects (Rogel et al, 2017; Saravia et al, 2020). Thus, we hypothesised that RIG-I negatively regulates CD8$^+$ T-cell anti-tumour function by inhibiting the AKT/glycolysis signalling pathway.

RIG-I is primarily expressed in immune cells such as T cells in the absence of viral infection (Jiang et al, 2023; Yang et al, 2017), but whether RIG-I plays an important role in adaptive immunity, particularly in CD8$^+$ T cells, remains poorly understood due to limited information. In this study, we showed that RIG-I serves as an intracellular checkpoint in the tumour microenvironment (TME). Using multiple solid tumour models, we found that overactivated T cells upregulate RIG-I to evade the killing of CD8$^+$ T cells via the AKT/glycolysis signalling pathway. Moreover, targeting RIG-I in human CD8$^+$ T cells enhances the effectiveness of adoptively transferred T cells in suppressing mouse and human tumour growth. Importantly, knocking out Rig-I in mouse CD8$^+$ T cells enhances intrinsic CD8$^+$ T-cell-mediated anti-tumour immunity and the response to anti-PD-1 therapy. Our findings identify RIG-I as a novel intracellular checkpoint and support the potential therapeutic value of targeting RIG-I in T cells.

## Results

### RIG-I is upregulated in tumour-infiltrating CD8$^+$ T cells and co-expressed with multiple immune checkpoints

We analyzed single-cell sequencing data from various tumour tissues, including hepatocellular carcinoma and intrahepatic cholangiocarcinoma (HCC and ICC), metastatic melanoma, head and neck squamous cell carcinoma (HNSCC), and colon cancer tissues, to identify novel markers associated with CD8$^+$ T-cell dysfunction in the TME (Cillo et al, 2020; Ma et al, 2021; Tirosh et al, 2016; Zhang et al, 2020). Our findings indicated that immune checkpoints (PDCD1, ENTPD1 and TIGIT), the T-cell exhaustion state-associated transcription factor TOX, and the gene DDX58 (transcriptional protein RIG-I) are co-expressed in tumour-infiltrating CD8$^+$ T cells (Figs. 1A–D; Fig. EV1A–D). Further analysis revealed that RIG-I, but not other RNA sensors [IFIH1 (MDA5) and DHX58 (LGP2)], the DNA sensor STING (STING), and the key downstream molecule of the RNA sensor MAVS (IPS-1), was more highly expressed in CD8$^+$ T cells within the TME than in other cells (Figs. 1E,F and EV1E,F). We conducted a detailed subpopulation analysis of infiltrating CD8$^+$ T cells in HCC and found that RIG-I was particularly expressed in exhausted cell types (LAG-3-CD8$^+$ T cells) (Fig. EV1G,H).

We used multiple immunohistochemistry (mIHC) to examine RIG-I expression in CD8$^+$ T cells and to confirm our findings. Our results revealed significantly higher RIG-I expression in tumour-infiltrating CD8$^+$ T cells was significantly higher in HCC and colon cancer tumour tissues than that in peritumoural tissues (Figs. 1G–I and EV2A–C). Additionally, Rig-I expression in tumour tissues was upregulated compared to that in CD8$^+$ T cells in the spleen of the Hepa1-6, MC38 and B16F10 mouse models (Fig. EV2D–F). Although single-cell sequencing analysis did not reveal a correlation between the expression of RIG-I in CD8$^+$ T cells infiltrating tumours and the expression of granzyme-B or IFN-γ (Fig. EV2G), in HCC and CRC tumours, infiltrating CD8$^+$ T cells exhibited secretion of granzyme-B compared to those in peritumoural tissues (Figs. 1J and EV2H). The above data suggest that RIG-I is specifically upregulated in tumour-infiltrating CD8$^+$ T cells and may function as a novel intracellular checkpoint that negatively regulates the anti-tumour activity of CD8$^+$ T cells.

### Rig-I negatively regulates the differentiation, development and anti-tumour activity of CD8$^+$ T cells ex vivo

The upregulation of RIG-I expression has been confirmed to be associated with the attenuation of anti-tumour cytokines in CD8$^+$ T cells (Fig. 1). We isolated CD8$^+$ T cells from the spleens of Rig-I$^{+/+}$

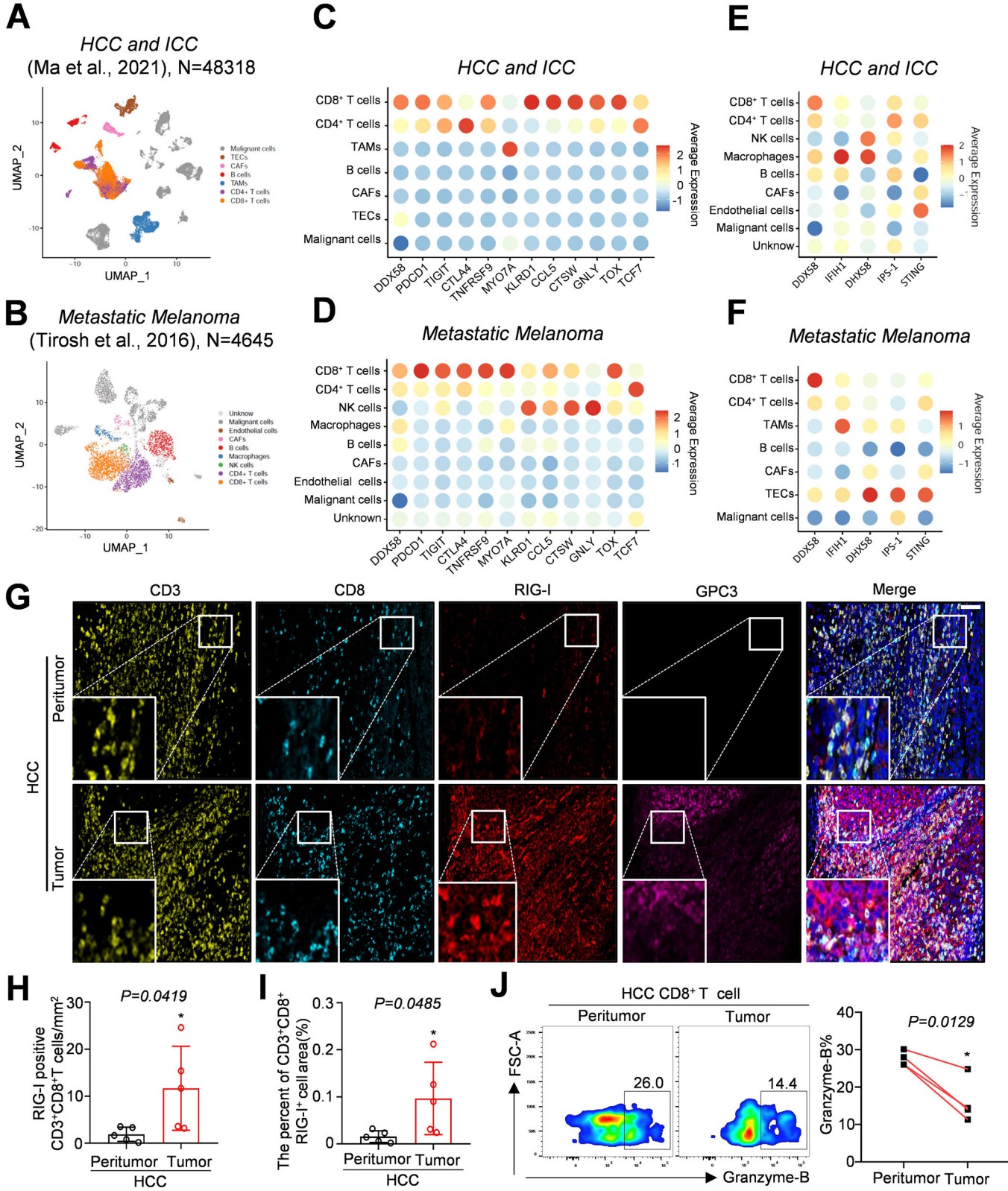

**Figure 1. Screening single-cell sequencing data and verification of RIG-I upregulation in CD8⁺ T cells infiltrating the TME.**

(A, B) Cluster analysis of cell populations in HCC, ICC and metastatic melanoma. (C, D) Relative expression levels of the *DDX58, PDCD1, TIGIT, CTLA-4,* and *TOX*-associated genes in patients with HCC, ICC (C), and metastatic melanoma (D). (E, F) Relative expression levels of *DDX58, IFIH1, DHX58, IPS-1,* and *STING* in HCC, ICC (E) and metastatic melanoma (F) tissues. (G–I) Multiple immunohistochemistry and statistical analyses of human HCC specimens; white scale bar = 50 μm. (J) The expression of granzyme-B secreted by infiltrating CD8⁺ T cells in the peritumor and tumour tissues of HCC by flow cytometry after stimulating with PMA/ionomycin and GolgiStop. Data information: The data represented different numbers ($n = 5$ or 4) of biological replicates and were shown as the means ± SEMs. Two-tailed unpaired Student's *t*-test was used in (H) ($P = 0.0419$) and (I) ($P = 0.0485$). Two-tailed paired Student's test was used in (J) ($P = 0.0129$). *$P < 0.05$, **$P < 0.01$, and ***$P < 0.001$ compared with peritumour tissues. Source data are available online for this figure.

and *Rig-I⁻ᐟ⁻* mice to perform RNA-seq analysis and enriched differentially expressed genes/signalling pathways were identified to confirm that RIG-I negatively regulates CD8⁺ T-cell function (Appendix Fig. S1; Dataset EV1). Knocking out *Rig-I* enhanced the enrichment of signalling pathways related to immune response regulation. Specifically, key genes related to leucocyte-mediated cytotoxicity, regulation of cell killing, and IFN-γ production were selected (Fig. 2A). The differentiation, development, and anti-tumour cytokines of CD8⁺ T cells were examined and evaluated using flow cytometry to further investigate the effect of *Rig-I* on CD8⁺ T cells. We found that knocking out *Rig-I* reduced the proportion of naïve CD8⁺ T cells, and increased the proportions of central memory and effector CD8⁺ T cells (Fig. 2B,C). In addition, knocking out *Rig-I* enhanced the secretion of anti-tumour cytokines such as CD107a, perforin, granzyme-B, IFN-γ, and TNF-α by CD8⁺ T cells (Fig. 2D–H). Collectively, these data provide strong evidence that *Rig-I* negatively regulates CD8⁺ T-cell function and that knocking out *Rig-I* enhances the differentiation, development, and anti-tumour activity of CD8⁺ T cells ex vivo.

## *Rig-I* knockout enhances the anti-tumour immunity of CD8⁺ T cells in vivo

We constructed a subcutaneous mouse HCC model (Hepa1-6), a melanoma mouse model (B16F10), and a colon cancer mouse model (MC38) in *Rig-I⁺ᐟ⁺* and *Rig-I⁻ᐟ⁻* C57BL/6J mice to further explore whether *Rig-I* knockout could improve the antitumor immunity of CD8⁺ T cells in vivo. We subcutaneously inoculated *Rig-I⁺ᐟ⁺* and *Rig-I⁻ᐟ⁻* mice with $3 \times 10^6$ Hepa1-6 cells, $1 \times 10^5$ B16F10 cells or $2 \times 10^5$ MC38 cells. The results indicated that *Rig-I⁻ᐟ⁻* mice had slower tumour growth than *Rig-I⁺ᐟ⁺* mice (Fig. 3A±C). The survival analysis results were consistent with the tumour growth results, showing that knocking out *Rig-I* could significantly prolong the survival time of tumour-bearing mice in the whole knockout and transfer models (Fig. EV3A,B). we intraperitoneally administered 250 μg control immunoglobulin (cIg) or neutralising antibodies against CD8a (anti-CD8a) to both *Rig-I⁺ᐟ⁺* and *Rig-I⁻ᐟ⁻* mice before and after subcutaneous inoculation with MC38 tumours to further investigate the role of CD8⁺ T cells in the enhanced anti-tumour immunity of *Rig-I* knockout mice, and the results revealed that *Rig-I⁻ᐟ⁻* CD8⁺ T cells play a crucial role in the anti-tumour function (Fig. 3D). We subcutaneously inoculated mice with MC38-OVA, a colon cancer cell line that overexpresses ovalbumin, which could activate CD8⁺ T cells in mice, to confirm that the strong anti-tumour phenotype of *Rig-I⁻ᐟ⁻* mice depends on the anti-tumour specificity of CD8⁺ T cells. As expected, knockout of *Rig-I* slowed the growth of MC38-OVA tumours in *Rig-I⁻ᐟ⁻* mice (Fig. 3E), suggesting that *Rig-I* was involved in the negative

regulation of the CD8⁺ T-cell anti-tumour function. An analysis of immune cells infiltration in the TME, revealed that compared to tumour-bearing *Rig-I⁺ᐟ⁺* mice, tumour-bearing *Rig-I⁻ᐟ⁻* mice exhibited a significantly greater proportion of CD8⁺ T cells, no significant change in the proportion of CD4⁺ T cells, a significantly greater CD8⁺/CD4⁺ T-cells ratio, and significantly higher IFN-γ levels in CD8⁺ or CD4⁺ T cells (Fig. 3F–K).

An analysis of splenic immune cells in mice with MC38 tumours revealed that CD8⁺ T cells in the spleens of *Rig-I⁻ᐟ⁻* mice exhibited increased IFN-γ secretion (Fig. EV3C–E). We conducted cytotoxicity experiments in vitro and adoptive transfer experiments in vivo to confirm the direct impact of knocking out *Rig-I* on CD8⁺ T cells while excluding its indirect effects on other immune cells. Knocking out *Rig-I* in CD8⁺ T cells from OT-1 mice and coculturing them with MC38-OVA cells at different target-effector ratios revealed that the absence of *Rig-I* enhanced the specific killing of tumour cells by CD8⁺ T cells (Fig. 4A). We transferred *Rig-I⁺ᐟ⁺* and *Rig-I⁻ᐟ⁻* CD8⁺ T cells into an MC38 tumour-bearing immunodeficient mouse model and confirmed that knocking out *Rig-I* slowed tumour growth, reduced the tumour volume and weight, increased the infiltration of CD8⁺ T cells, and enhanced the secretion of the anti-tumour cytokine granzyme-B (Fig. 4B–I). Collectively, these data demonstrate that knocking out *Rig-I* in CD8⁺ T cells increased the infiltration of CD8⁺ T cells and improved their anti-tumour function. Further elucidation of the mechanism by which *Rig-I* negatively regulates the anti-tumour function of CD8⁺ T cells in the TME will provide a theoretical basis for tumour immunotherapy targeting *Rig-I*.

## The T-cell activation-mediated *Rig-I*/AKT/glycolysis signalling pathway negatively regulates the anti-tumour function of CD8⁺ T cells in the TME

Overactivation of T cells infiltrated in the TME is a key factor leading to T-cell exhaustion, but the mechanism of CD8⁺ T-cell exhaustion caused by RIG-I remains unclear. We sorted naïve CD8⁺ T cells from the spleens of wild-type mice, stimulated them with a CD28 antibody and gradient concentrations of a CD3 antibody for 72 h to mimic the stimulation of the T-cell receptor (TCR) by tumour antigens, and detected *Rig-I* expression using Western blotting to investigate the upregulation of *Rig-I* induced by the activation of T cells. *Rig-I* was obviously upregulated as the concentration of the CD3 antibody increased (Fig. 5A), and PD-1 expression was accompanied by a significant increase in *Rig-I* expression (Fig. 5B). In addition, the late activation marker CD25 and the secretion of anti-tumour cytokines such as CD107a and IFN-γ were significantly higher in *Rig-I*-knockout CD8⁺ T cells than in wild-type *Rig-I* CD8⁺ T cells (Fig. 5C–E). These data indicate that *Rig-I* can significantly enhance PD-1 expression, reduce the expression of activation markers and reduce the release of

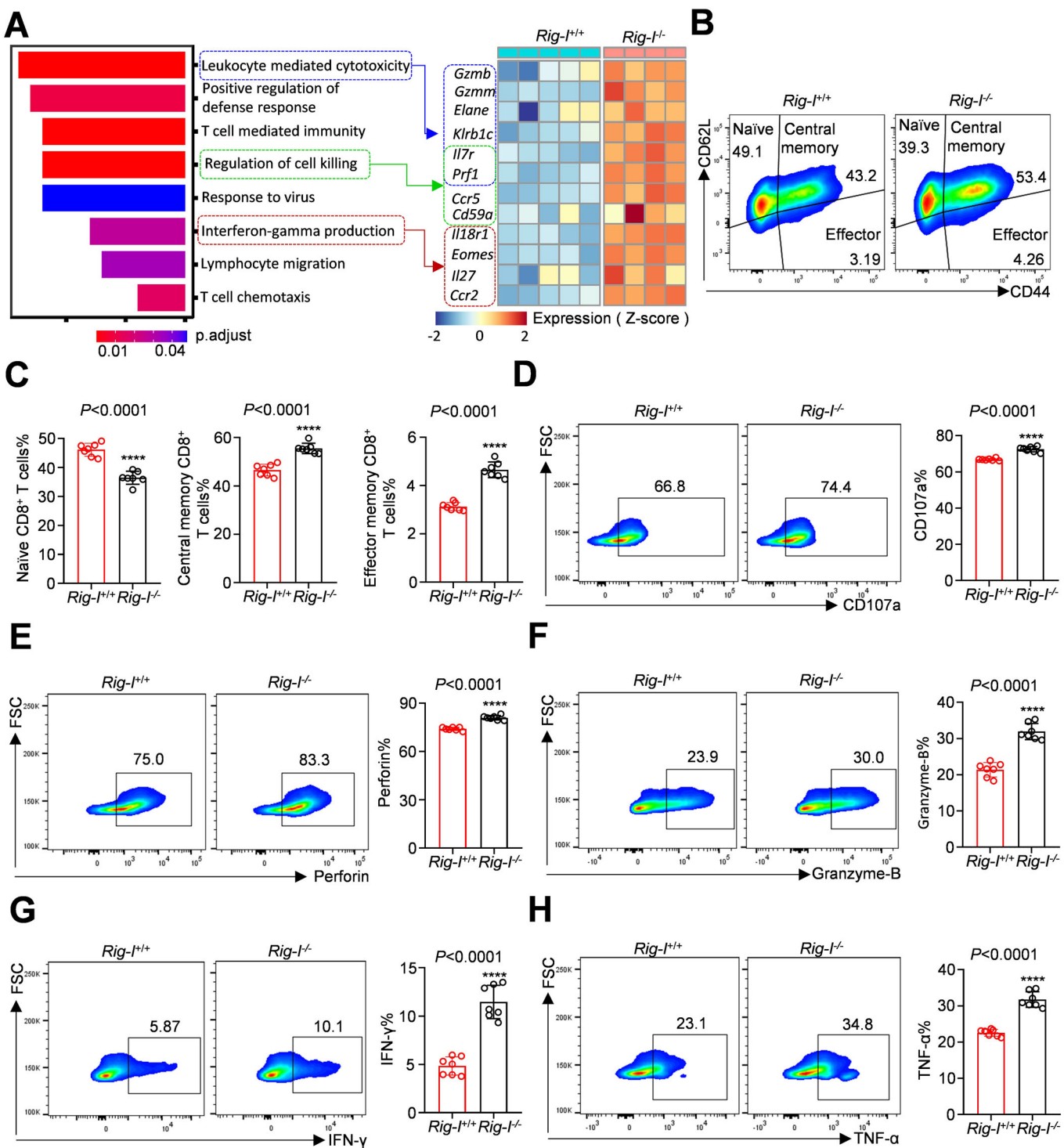

**Figure 2. *Rig-I* knockout enhanced the anti-tumour function of CD8⁺ T cells purified from the spleen ex vivo.**

(**A**) RNA-seq and KEGG enrichment analysis of signalling pathways (left panel) and key genes associated with leucocyte-mediated cytotoxicity, regulation of cell killing and interferon-gamma production (right panel) in CD8⁺ T cells from *Rig-I⁺/⁺/Rig-I⁻/⁻* mouse spleens. (**B, C**) Flow cytometry analysis of the development (including naïve, central memory and effector cells) of CD8⁺ T cells purified from *Rig-I⁺/⁺/Rig-I⁻/⁻* mouse spleens after α-CD3 and α-CD28 activation ex vivo. (**D–H**) Flow cytometry analysis of CD107a, perforin, granzyme-B, IFN-γ and TNF-α levels in CD8⁺ T cells purified from *Rig-I⁺/⁺/Rig-I⁻/⁻* mouse spleens after α-CD3 and α-CD28 activation and stimulation with PMA/ionomycin and GolgiStop ex vivo. Data information: A hypergeometric test was used in (**A**). The data represented different numbers ($n = 7$) of biological replicates and were shown as the means ± SEMs in (**C–H**). Two-tailed unpaired Student's test was used in (**C–H**) ($P < 0.0001$). ****$P < 0.0001$ compared with the *Rig-I⁺/⁺* group. Source data are available online for this figure.

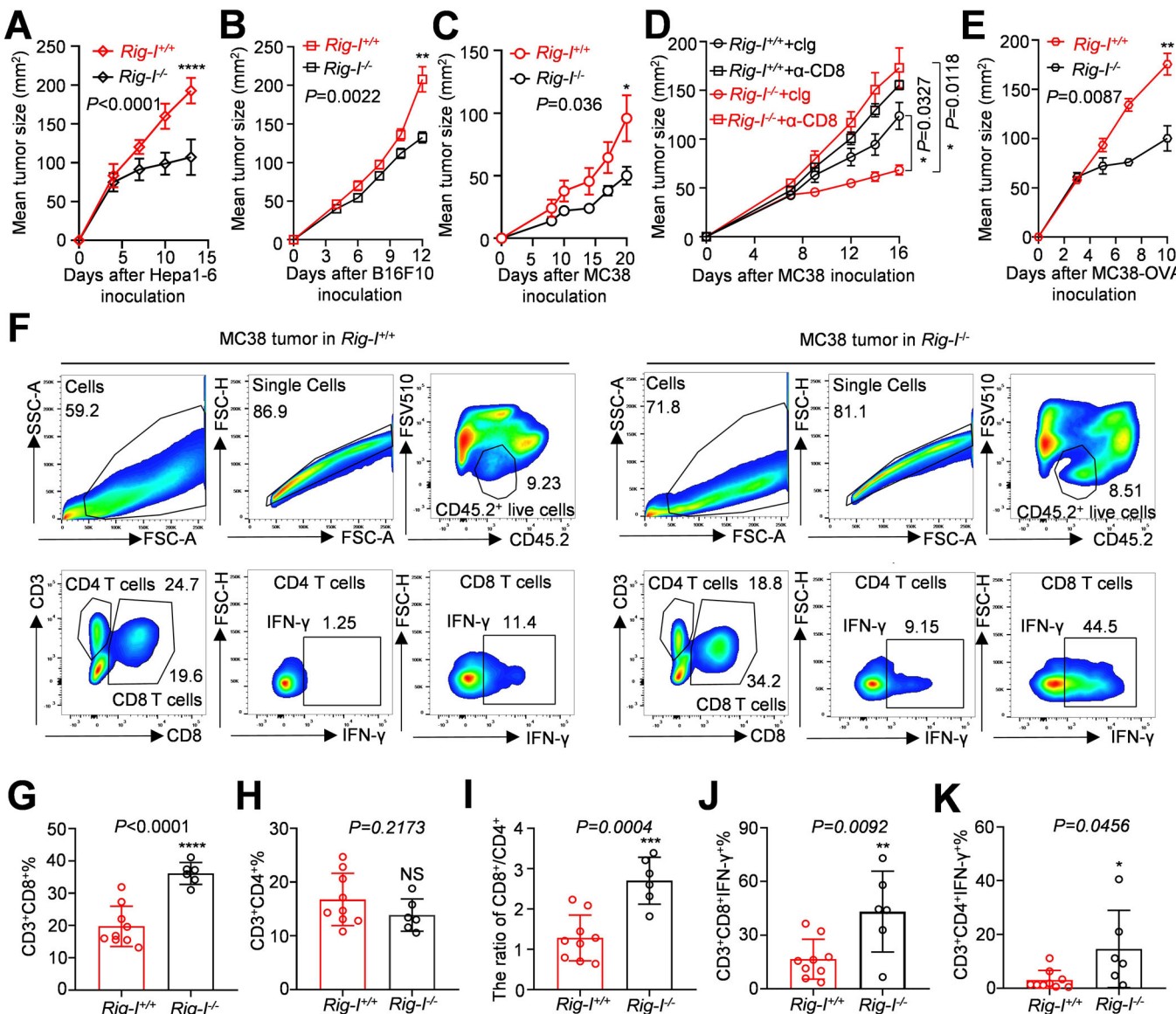

**Figure 3.** *Rig-I* knockout enhanced the anti-tumour function of CD8⁺ T cells in animal models.

(A–C) Hepa1-6, MC38 and B16F10 cells were inoculated subcutaneously into *Rig-I*⁺/⁺ and *Rig-I*⁻/⁻ mice, and the growth curves of Hepa1-6, MC38 and B16F10 tumours are shown. (D) After mice were treated with control immunoglobulin or a CD8 neutralising antibody, MC38 cells were inoculated subcutaneously into *Rig-I*⁺/⁺ and *Rig-I*⁻/⁻mice, and tumour growth curves are shown. (E) MC38-OVA cells were inoculated subcutaneously into *Rig-I*⁺/⁺ and *Rig-I*⁻/⁻ mice, and the growth curves of MC38-OVA tumours were shown. (F) Flow cytometry analysis of the IFN-γ level and proportion of CD8⁺ T/CD4⁺ T cells infiltrating tumours formed from the colon cancer cell line MC38 in *Rig-I*⁺/⁺ and *Rig-I*⁻/⁻ mice after stimulation with PMA/ionomycin and GolgiStop. (G–I) Statistics of the proportions of CD8⁺ T (G) and CD4⁺ T cells (H) and the ratio of CD8⁺ T/CD4⁺ T cells (I). (J, K) Statistics of IFN-γ levels in CD8⁺ T (J) and CD4⁺ T cells (K) after stimulation with PMA/ionomycin and GolgiStop for 4 h. Data information: The data represented different numbers ($n = 6$–9) of biological replicates and were shown as the means ± SEMs. Two-tailed unpaired Student's t-test was used in (A) ($P < 0.0001$), (G) ($P < 0.0001$), (H) ($P = 0.2173$) and (J) ($P = 0.0092$). Two-way ANOVA was used in (D) (Exact p values were reported on graphs). A two-tailed Mann–Whitney U-test was used in (B) ($P = 0.0022$), C ($P = 0.036$), (E) ($P = 0.0087$) and (K) ($P = 0.0456$). *$P < 0.05$, **$P < 0.01$, ***$P < 0.001$, ****$P < 0.0001$, and NS not significant compared with the *Rig-I*⁺/⁺ group. Source data are available online for this figure.

anti-tumour cytokines due to activation, attenuating the anti-tumour function of CD8⁺ T cells PD-1.

Inhibition of AKT phosphorylation is a key event in the downstream signalling of RIG-I, and activation of the AKT signalling pathway is the key to maintaining the anti-tumour function of CD8⁺ T cells via energy metabolism. We isolated, purified and activated naïve CD8⁺ T cells from the spleens of *Rig-*

*I*⁺/⁺ and *Rig-I*⁻/⁻ mice treated with α-CD3 and α-CD28 and then detected the total AKT and phosphorylated AKT (p-AKT Th308) protein levels using Western blotting to verify that RIG-I inhibits AKT phosphorylation in CD8⁺ T cells. As expected, p-AKT was activated in *Rig-I*⁻/⁻ CD8⁺ T cells (Fig. 5F). Similarly, significantly increased p-AKT levels were also observed in human CD8⁺ T cells compared with mouse CD8⁺ T cells (Fig. 5G,H). We treated

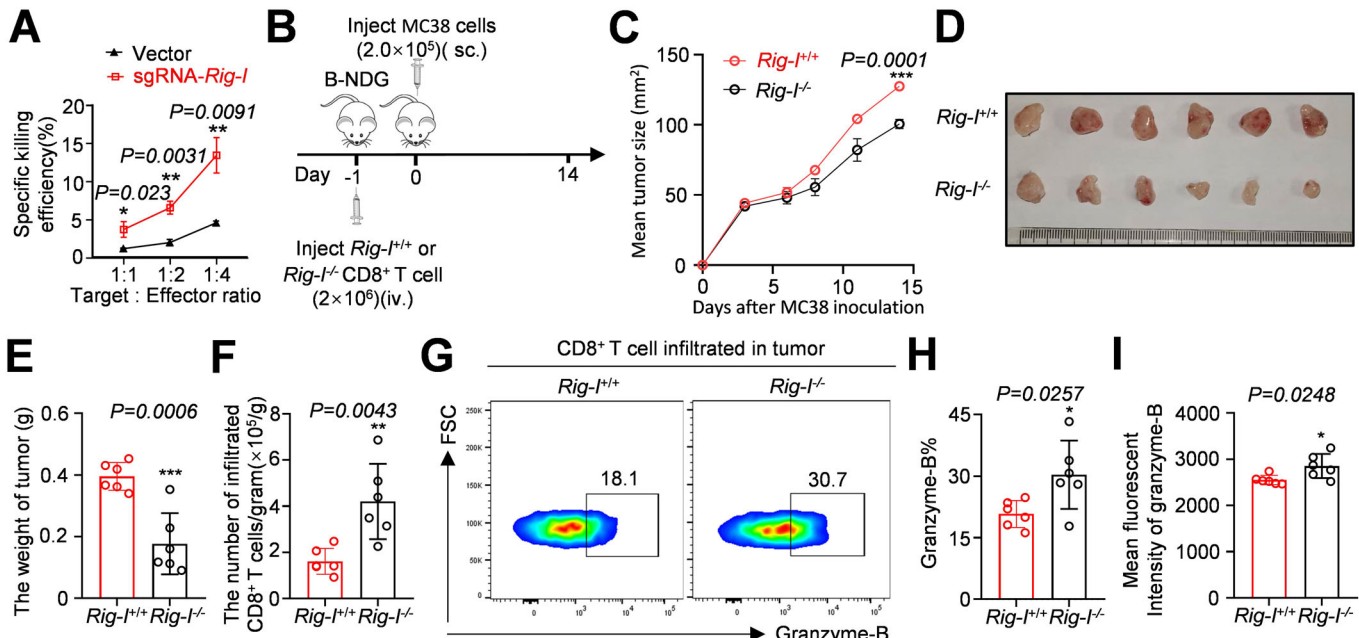

**Figure 4. Knocking out *Rig-I* enhanced the anti-tumour function of CD8⁺ T cells in a transfer experiment.**

(A) Coculture of *Rig-I* knockout CD8⁺ T cells from OT-1 mice with MC38-OVA cells at different ratios to determine the specific killing efficiency. (B) Flowchart of the transfer of *Rig-I$^{+/+}$* and *Rig-I$^{-/-}$* mouse spleen-derived CD8⁺ T cells for the treatment of MC38 tumours. (C–E) Analysis of tumour growth curves, tumour sizes and tumour weights. (F–I) Flow cytometry analysis of the absolute number of CD8⁺ T cells infiltrating tumours, the secretion of granzyme-B and the mean fluorescence intensity after stimulation with PMA/ionomycin and GolgiStop. Data information: The data represented different numbers ($n = 4$ or 6) of biological replicates and were shown as the means ± SEMs. Two-tailed unpaired Student's test was used in (A) (Exact $p$ values were reported on graphs), (C) ($P = 0.0001$), (E) ($P = 0.0006$), (F) ($P = 0.0043$), (H) ($P = 0.0257$) and (I) ($P = 0.0248$). *$P < 0.05$, **$P < 0.01$ and ***$P < 0.001$ compared with the vector or *Rig-I$^{+/+}$* group. Source data are available online for this figure.

purified CD8⁺ T cells from *Rig-I$^{+/+}$* and *Rig-I$^{-/-}$* mice with PI3K, AKT and glycolysis inhibitors for 3 days and evaluated the expression of CD69, CD25, CD107a and IFN-γ using flow cytometry to further confirm that *Rig-I* negatively regulates the anti-tumour function of CD8⁺ T cells through the AKT/glycolysis signalling pathway (Figs. 5I–L and EV4A–F). We found that these inhibitors inhibited the expression of CD69, CD25, CD107a and IFN-γ in *Rig-I$^{+/+}$* and *Rig-I$^{-/-}$* CD8⁺ cells. However, *Rig-I$^{-/-}$* CD8⁺ T cells were more sensitive to inhibitors of PI3K, AKT, and glycolysis, indicating that *Rig-I* negatively regulates the anti-tumour activity of CD8⁺ T cells through the PI3K/AKT/glycolysis signalling pathway.

Collectively, these data indicate that TCR signalling in the TME can induce *Rig-I* upregulation in CD8⁺ T cells, which inhibits the AKT/glycolysis signalling pathway to negatively regulate the anti-tumour activity of CD8⁺ T cells. Hence, targeting the *Rig-I*/AKT/glycolysis signalling pathway could partially restore the anti-tumour function of CD8⁺ T cells in vitro.

## Targeting RIG-I in human CD8⁺ T cells enhances anti-tumour activity

We observed that knocking out *Rig-I* in mouse CD8⁺ T cells improved their killing ability and slowed tumour progression in multiple mouse tumour models (Fig. 3). We generated immortalised lymphoblastoid cell lines (LCLs) by infecting human PBMCs with Epstein–Barr virus (EBV) for 21 days to further explore

whether targeting RIG-I could also enhance the anti-tumour function of human CD8⁺ T cells. Then, EBV-specific cytotoxic T lymphocytes (CTLs) were amplified in vitro by coculturing PBMCs from the same donor and irradiated LCLs (80 Gy) as feeder cells, ensuring that the TCR matched the LCLs' HLA-I. Finally, we used lentivirus-delivered sgRNA (Fig. 5G) to knock out RIG-I in EBV-specific CTLs, achieving more than 80% knockout efficiency. After subcutaneous inoculation of $1 \times 10^7$ LCLs in B-NDG mice, $9 \times 10^6$ RIG-I-knockout EBV-specific CD8⁺ T cells were intravenously administered on days 4, 8 and 12 (Fig. 6A). Adoptive transfer of RIG-I-knockout CTLs slowed tumour growth and reduced the tumour volume and weight (Fig. 6B–D). An analysis of CD8⁺ T cells in subcutaneous LCL tumours revealed that the secretion of the anti-tumour cytokines IFN-γ, granzyme-B and perforin was significantly increased in the RIG-I knockout group compared to the control group (Fig. 6E–G). Our findings showed that RIG-I knockout in human CD8⁺ T cells also slowed tumour growth and enhanced the function of CD8⁺ T cells in vivo.

## Synergistic control of tumour progression by combining *Rig-I* targeting and PD-1 blockade

In cancer treatment, the use of a single drug often leads to drug resistance, and combining different drugs can overcome this resistance, enhance drug efficacy, and broaden the scope of treatment options. We aimed to explore whether targeting *Rig-I* in combination with an anti-PD-1 antibody could increase the

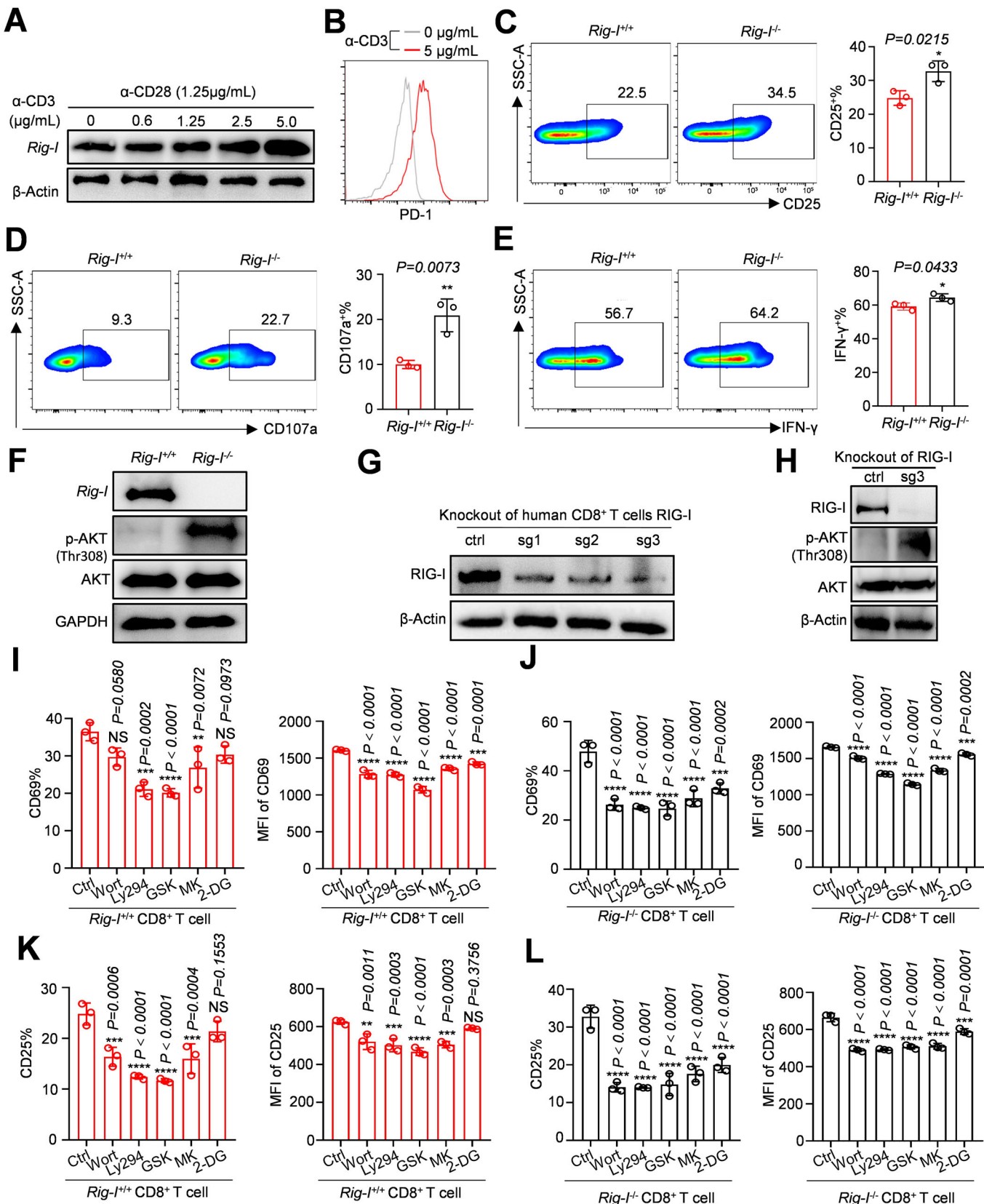

◄ **Figure 5. The activation of CD8⁺ T cells induced the upregulation of *Rig-I*, inhibiting the PI3K/AKT/glycolysis signalling pathway to counteract its anti-tumour function.**

(**A–E**) Naïve CD8⁺ T cells were isolated from the spleens of wild-type mice, and after 72 h of costimulation with α-CD3/α-CD28, *Rig-I* expression was detected using Western blotting (**A**), and the expression of PD-1 (**B**), CD25 (**C**), CD107a (**D**) and IFN-γ (**E**) in CD8⁺ T cells was detected using flow cytometry after stimulation with PMA/ ionomycin and GolgiStop. (**F**) After 72 h of costimulation with α-CD3/α-CD28, the expression of RIG-I, p-AKT (Thr308), AKT, and GAPDH in WT or KO *Rig-I* CD8⁺ T cells from mouse spleens was detected using Western blotting. (**G**) Validation of the efficiency of RIG-I knockout in human CD8⁺ T cells. (**H**) After knocking out RIG-I, the expression of RIG-I, p-AKT (Thr308), AKT, and β-Actin in human CD8⁺ T cells was detected using Western blotting. (**I–L**) Naïve CD8⁺ T cells from *Rig-I⁺/⁺*, *Rig-I⁻/⁻* mouse spleens were treated with PI3K, AKT and glycolysis inhibitors after α-CD3/α-CD28 stimulation, and the proportions and mean fluorescence intensities of CD69 (**I–J**) and CD25 (**K–L**) in CD8⁺ T cells were detected using flow cytometry. Data information: The data represented different numbers (*n* = 3) of biological replicates and were shown as the means ± SEMs. Two-tailed unpaired Student's test was used in (**C**) (*P* = 0.0215), (**D**) (*P* = 0.0073) and (**E**) (*P* = 0.0433). One-way multiple comparisons ANOVA with Tukey's correction test was used in (**I–L**) (Exact *p* values were reported on graphs). **P* < 0.05, ***P* < 0.01, ****P* < 0.001, *****P* < 0.0001 and NS not significant compared with the *Rig-I⁺/⁺* or Ctrl group. Ctrl control, Wort wortmannin, Ly294 Ly294002, GSK GSK690693, 2-DG 2-deoxy-ᴅ-glucose. Source data are available online for this figure.

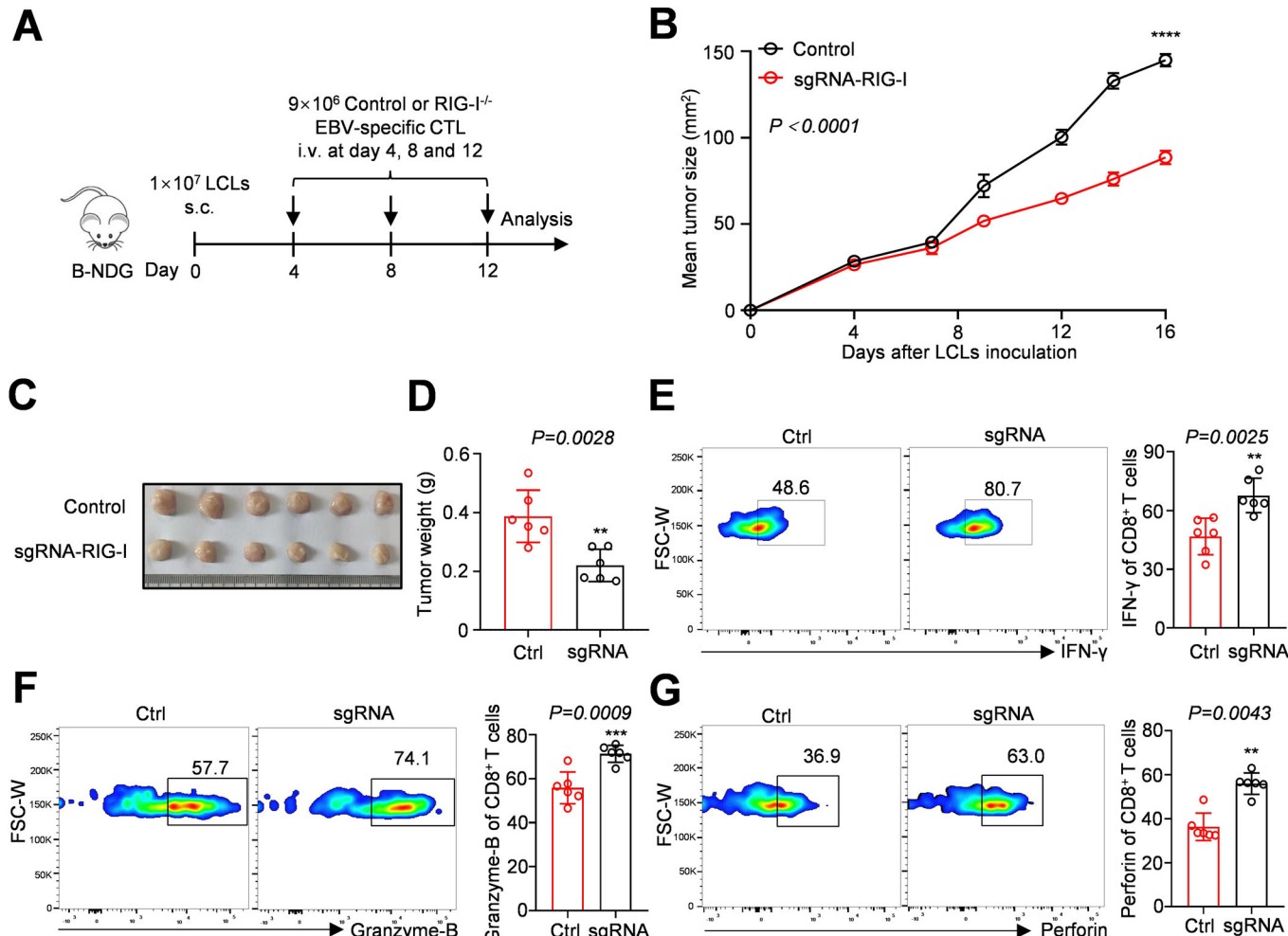

**Figure 6. Adoptive cell therapy targeting RIG-I in EBV-specific CTL cells enhanced its anti-tumour function in the LCL model.**

(**A, B**) Schematic of adoptive cell therapy targeting RIG-I in an EBV-specific CTL tumour model and growth curves of LCL tumours. (**C, D**) The size and weight of the LCL tumours. (**E–G**) Flow cytometry analysis of the proportions of IFN-γ, granzyme-B and perforin in the control and RIG-I knockout groups after stimulation with PMA/ ionomycin and GolgiStop. Data information: The data represented different numbers (*n* = 6) of biological replicates and were shown as the means ± SEMs. Two-tailed unpaired Student's *t*-test was used in (**B**) (*P* < 0.0001), (**D**) (*P* = 0.0028), (**E**) (*P* = 0.0025), (**F**) (*P* = 0.0009). A two-tailed Mann–Whitney *U*-test was used in (**G**) (*P* = 0.0043). ***P* < 0.01, ****P* < 0.001 and *****P* < 0.0001 compared with the control group. Source data are available online for this figure.

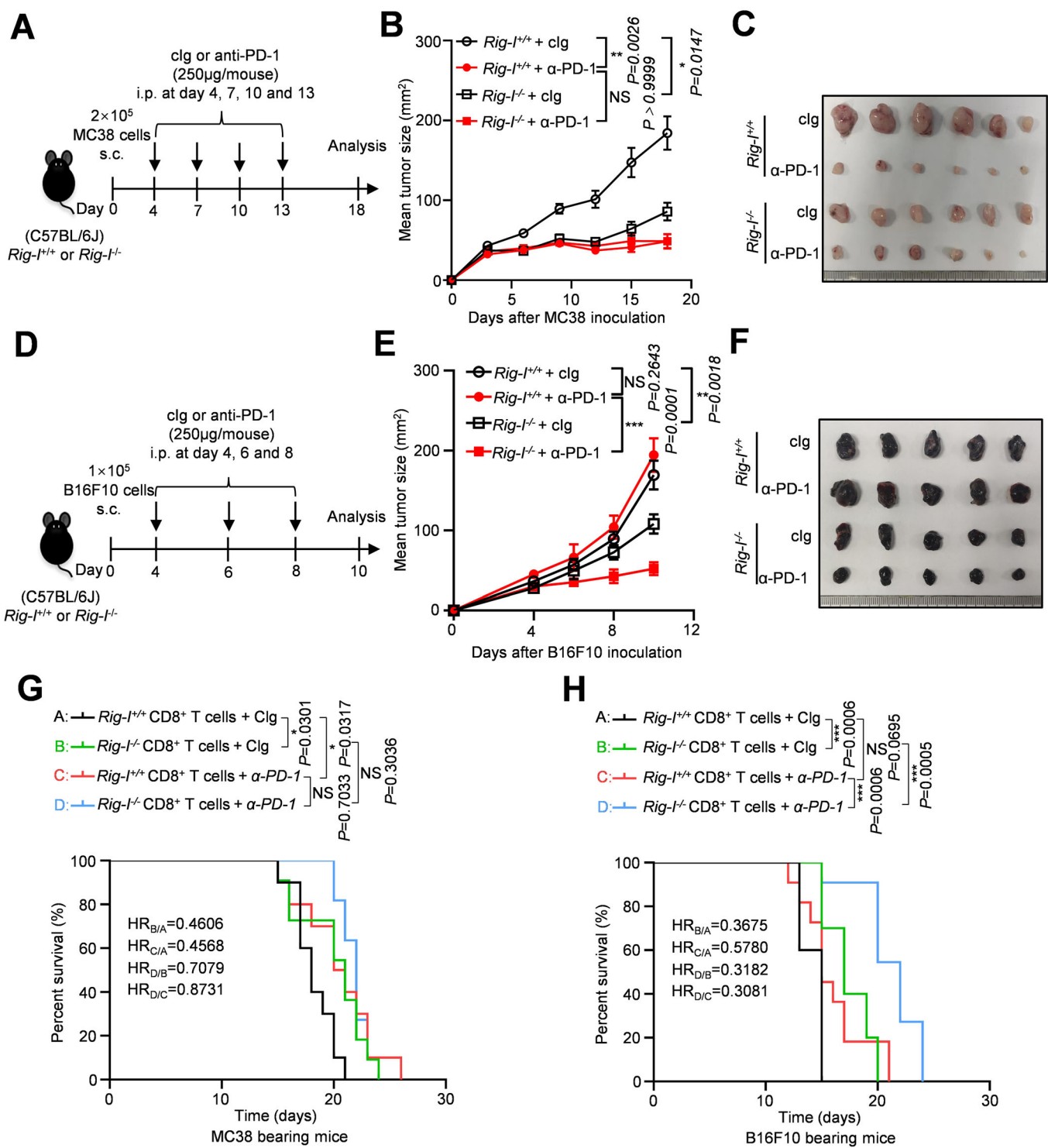

sensitivity of PD-1 antibody-insensitive tumours. First, we subcutaneously inoculated $2 \times 10^5$ MC38 colon cancer cells, which are tumour cells that are sensitive to anti-PD-1 antibody treatment, into $Rig\text{-}I^{+/+}$ and $Rig\text{-}I^{-/-}$ mice and intraperitoneally injected 250 μg of cIg or anti-PD-1 antibody into each mouse on days 4, 7, 10 and 13 (Fig. 7A). We found that the PD-1 antibody alone or $Rig\text{-}I$ knockout significantly inhibited tumour growth, but the PD-1

antibody combined with $Rig\text{-}I$ knockout did not have a synergistic effect (Fig. 7B,C).

Next, we subcutaneously inoculated $1 \times 10^5$ B16F10 melanoma cells, which are tumour cells that are insensitive to PD-1 antibody treatment, into $Rig\text{-}I^{+/+}$ and $Rig\text{-}I^{-/-}$ mice and intraperitoneally injected 250 μg of cIg or PD-1 antibody into each mouse on days 4, 6 and 8 (Fig. 7D). We found that the PD-1 antibody alone did not

◄ **Figure 7.** *Rig-I* knockout combined with PD-1 monoclonal antibodies can be used to treat solid tumours that are insensitive to PD-1 antibodies.

(A) *Rig-I*$^{+/+}$ and *Rig-I*$^{-/-}$ mice were inoculated subcutaneously with MC38 cells and treated with a control immunoglobulin (cIg) or PD-1 monoclonal antibody, respectively. (B) Growth curve of MC38 tumours. (C) Anatomy of MC38 tumours. (D) *Rig-I*$^{+/+}$ and *Rig-I*$^{-/-}$ mice were inoculated subcutaneously with B16F10 cells and treated with cIg or a PD-1 monoclonal antibody, respectively. (E) Growth curve of B16F10 tumours. (F) Anatomy of B16F10 tumours. (G, H) Survival curves of immunodeficient mice with MC38 (G) and B16F10 (H) tumours in which *Rig-I*$^{+/+}$ or *Rig-I*$^{-/-}$ naïve CD8$^+$ T cells were transferred alone, treated with anti-PD-1 antibodies alone, or treated with their combination. Data information: The data represented different numbers ($n = 5$ or 6 for **B** and **E**, $n = 10$ or 11 for **G** and **H**) of biological replicates, and were shown as the means ± SEMs. Two-way ANOVA was used in (**B**, **E**) (Exact *p* values were reported on graphs). The log-rank test was used in (**G**, **H**) (Exact *p* values were reported on graphs). *$P < 0.05$, **$P < 0.01$, ***$P < 0.001$, and NS not significant compared with the other groups. Source data are available online for this figure.

inhibit the growth of B16F10 tumours, while *Rig-I* knockout slightly inhibited tumour growth (Fig. 7E,F). Interestingly, *Rig-I* knockout combined with PD-1 antibody treatment enhanced the sensitivity of B16F10 tumours to immunotherapy (Fig. 7E,F). Furthermore, in the abovementioned PD-1-sensitive and PD-1-nonsensitive models, the effect of the combined therapy on survival was further validated through the adoptive transfer of *Rig-I*-knockout CD8$^+$ T cells in combination with treatment with PD-1 antibodies. In the PD-1-sensitive MC38 cell model, knocking out *Rig-I* alone or administering the PD-1 antibody improved survival, but the combination of both did not (Fig. 7G). In the PD-1-insensitive B16F10 cell model, the administration of an anti-PD-1 antibody alone did not improve survival, while knocking out *Rig-I* alone improved survival, and the combination of both had a synergistic effect (Fig. 7H). These findings indicate that knocking out *Rig-I* or administering an anti-PD-1 antibody alone cannot inhibit the growth of PD-1 antibody-sensitive tumours. However, combining *Rig-I* knockout with an anti-PD-1 antibody can increase the susceptibility of PD-1 antibody-insensitive tumours to immunotherapy. Therefore, knocking out *Rig-I* in combination with an anti-PD-1 antibody may represent a promising approach for treating PD-1 antibody-insensitive tumours.

## Discussion

Immune checkpoint inhibitors (ICIs), such as PD-1 and CTLA-4 inhibitors, have shown great efficacy in the treatment of various solid tumours, but not all patients respond to treatment. Therefore, a need exists to understand the fundamentals of tumour immune escape and explore new immune checkpoints. This study revealed RIG-I as a novel intracellular immune checkpoint that negatively regulates the anti-tumour activity of CD8$^+$ T cells. This study demonstrated that knocking out RIG-I enhances CD8$^+$ T-cell differentiation, development, and anti-tumour activity, thereby inhibiting tumour growth. RIG-I is upregulated in tumour-infiltrating CD8$^+$ T cells by factors such as TCR activation in the TME. Mechanistically, RIG-I negatively regulates the anti-tumour activity of CD8$^+$ T cells by inhibiting AKT phosphorylation. Furthermore, knocking out RIG-I also enhances the anti-tumour activity of CD8$^+$ T cells in a humanised tumour model. Importantly, combining knock-out RIG-I with an anti-PD-1 antibody enhances therapeutic efficacy in tumours insensitive to PD-1 antibodies.

Our results showed that RIG-I is upregulated in tumour-infiltrating CD8$^+$ T cells from humans and mice. These results suggest that RIG-I may act as a novel intracellular checkpoint to negatively regulate the anti-tumour activity of CD8$^+$ T cells. Consistent with a recent report, RIG-I expression in mice was upregulated in CD8$^+$ T cells that

infiltrated the colon compared to the expression in CD8$^+$ T cells that infiltrated the spleen (Jiang et al, 2023). RIG-I, a multifunctional molecule, is widely expressed in tumour cells, fibroblasts, and immune cells in the TME, with the highest expression in CD8$^+$ T cells. Considering that RIG-I is well known as a classic pattern recognition receptor for RNA viruses, several RIG-I agonists have been successfully developed for the treatment of solid tumours based on the role of RIG-I in activating innate immunity (Elion et al, 2018; Ellermeier et al, 2013; Poeck et al, 2008). The anti-tumour mechanism of RIG-I agonists is mainly activated by the MAVS/Type I IFN signalling pathway in tumour cells or innate immune cells (Elion et al, 2018; Heidegger et al, 2019a; Heidegger et al, 2019b). However, the function of RIG-I in adaptive immune cells, especially CD8$^+$ T cells, remains to be elucidated.

Our results confirmed that RIG-I negatively regulates the differentiation, development, and anti-tumour activity of CD8$^+$ T cells both in vitro and in vivo. Interestingly, RIG-I is a versatile effector molecule in different cell types. RIG-I is undoubtedly a significant molecule involved in innate immunity, and many excellent studies have examined the regulation of RIG-I expression in different cells. Our group has accumulated a wealth of experience in previous studies on RIG-I, such as RIG-I inhibiting the phosphorylation of STAT3 and AKT, which regulate immune homoeostasis and suppress leukaemia (Li et al, 2014; Yang et al, 2017). Furthermore, RIG-I activates the STAT1 molecule, to inhibit the proliferation ability of leukaemia cells (Jiang et al, 2011). RIG-I can regulate cell function through multiple molecular signals. Abnormal activation of STAT5 and AKT is closely related to tumour growth, proliferation, metastasis, and resistance, and more importantly, it is associated with the maintenance, expansion, and energy metabolism of immune cells. The work of Jiang et al and our study confirmed that the activation of TME-infiltrating CD8$^+$ T cells induces abnormal RIG-I expression, thereby inhibiting the activation of STAT5 and negatively regulating the phosphorylation of AKT at position 308, which in turn suppresses the anti-tumour function of CD8$^+$ T cells. Together with our study, Jiang's research confirmed that abnormal RIG-I expression is induced upon the activation of CD8$^+$ T cells infiltrating the TME (Jiang et al, 2023). However, the mechanisms differ: one involves inhibiting the activation of STAT5, while the other involves suppressing the phosphorylation of the 308th amino acid in AKT, thereby inhibiting the anti-tumour function of CD8$^+$ T cells. These two different mechanisms validate the same phenomenon, providing new insights for targeted tumour immunotherapy focusing on RIG-I molecules in CD8$^+$ T cells.

During the development and progression of cancer, tumours use various mechanisms to evade immune surveillance and suppress anti-tumour immune responses (Chen et al, 2023; Duan et al, 2022; Dubrot et al, 2022; Freed-Pastor et al, 2021). However, the regulatory mechanism of RIG-I in CD8$^+$ T cells remains unresolved. Our initial

exploration focused on TCR signalling-induced T-cell activation, with detailed mechanisms requiring further investigation, possibly due to the loss of KLHL22 leading to reduced RIG-I degradation (Zhou et al, 2020), or regulated by the NFAT transcription factor family in the process of CD8+ T-cell exhaustion that leads to RIG-I upregulation (Martinez et al, 2015; Tillé et al, 2023). The mechanism of RIG-I upregulation in activated CD8+ T cells may be diverse, and in future research, the key transcription factors and mechanisms that induce RIG-I upregulation in CD8+ T cells need to be investigated.

In addition to the potential of enhancing endogenous anti-tumour immunity and improving the response to anti-PD-1 therapy, our studies also confirmed the therapeutic potential of targeting RIG-I in adoptive T-cell immunotherapy, especially in epitope-specific T cells of human origin. Interestingly, the activation of the DNA sensor STING promotes the stemness of CD8+ T cells and enhances the anti-tumour function, in contrast to the activity of the RNA sensor RIG-I which negatively regulates the CD8+ T-cell anti-tumour function. While both RIG-I and STING can participate in regulating the function of CD8+ T cells in tumour immunity, their functions are completely opposite and involve different mechanisms. The cGAS-STING pathway utilises its own DNA sensing function to perceive enriched genomic DNA in CD8+ T cells, initiating the activation of the cGAS-STING innate immune signalling pathway to prevent excessive activation of CD8+ T cells, enhancing their stemness and anti-tumour function (Li et al, 2020). On the other hand, RIG-I does not exert immunoregulatory effects on CD8+ T cells through its own RNA sensing function. The study by Jiang et al and our research confirmed that tumour antigen stimulation of TCR in the TME leads to the upregulation of RIG-I, which negatively regulates the anti-tumour function of CD8+ T cells (Jiang et al, 2023). This result represents a significant discovery in recent years in the exploration of new functions of classic innate immune molecules in adaptive immune CD8+ T cells, providing a new direction for future T-cell immunotherapy targeting RIG-I in combination with cGAS-STING agonists. In the current and future studies of engineered cell therapy, such as CAR-T-cell or TCR-T-cell therapy, the knockout of one or more genes to increase the infiltration of CD8+ T cells in the TME, the killing effect on the tumour, and the persistence of immune cells are very useful tools and strategies (Mai et al, 2023; Prinzing et al, 2021; Wei et al, 2019). Additionally, we investigated innate immune cells such as NK cells and NK-like T cells. We found that *Rig-I* has a strong immunosuppressive effect on innate immune NK-like T cells (Appendix Fig. S2A–D), but not in NK cells (Appendix Fig. S2E–H). Therefore, our findings provide new possibilities for optimising engineered immune cell-based (T or NK-like T cells) therapy in the future.

Currently, PD-1 therapeutic antibodies are the most widely used immune checkpoint inhibitors in clinical practice. Nevertheless, the responsiveness of tumours to PD-1 antibodies is 10–20% (Morad et al, 2021). Notably, our research revealed an interesting finding: RIG-I knockout increases the proportion of CD8+ T cells that infiltrate the TME and enhances the killing activity of CD8+ T cells. Specifically, PD-1-sensitive tumours, which are infiltrated by many lymphocytes and upregulate PD-1, are the basis for the effectiveness of PD-1 antibodies. According to this model, the infiltration and functional activity of CD8+ T cells may have reached their maximum limits. This saturation state may be one of the reasons why RIG-I knockout did not enhance the effectiveness of therapy. In contrast, for tumours insensitive to PD-1, limited lymphocyte infiltration, insignificant PD-1 expression, and poor functional activity restrict the anti-tumour immune response of CD8+ T cells.

RIG-I knockout can increase the infiltration of CD8+ T cells, enhance PD-1 expression, and strengthen their anti-tumour function. This finding may be a key reason why knocking out RIG-I has a significant effect on PD-1-insensitive tumours.

In summary, we revealed that the activation of a novel intracellular checkpoint, RIG-I, impedes the anti-tumour activity of CD8+ T cells in various solid tumours. The upregulation of RIG-I through the AKT/glycolysis signalling pathway restrains the anti-tumour effect of CD8+ T-cell infiltrating in the TME. Furthermore, targeting RIG-I in CD8+ T cells suppresses tumour growth. As a novel intracellular checkpoint for CD8+ T cells in the TME, RIG-I is a promising therapeutic target for enhancing CD8+ T-cell function for the treatment of solid tumours, particularly those that are unresponsive to PD-1 antibodies.

# Methods

### Reagents and tools table

| Reagent/Resource | Reference or source | Identifier or catalogue number |
| --- | --- | --- |
| **Experimental Models** | | |
| MC38 cells (M. sapiens) | ATCC | CVCL_B288 |
| Hepa1-6 cells (M. sapiens) | ATCC | CRL-1830 |
| MC38-OVA cells (M. sapiens) | ATCC | CVCL_XJ96 |
| B16F10 cells (M. sapiens) | ATCC | CVCL_0159 |
| HEK293T cells (H. sapiens) | ATCC | CVCL_0063 |
| C57BL6/J (M. musculus) | Zhuhai Bestest Biotechnology | N/A |
| B-NDG | Zhuhai Bestest Biotechnology | N/A |
| **Recombinant DNA** | | |
| plentiCRISPR v2 | Addgene | 52961-DNA.cg |
| psPAX2 | Addgene | 12260 |
| pMD2.G | Addgene | 12259 |
| **Antibodies** | | |
| BV510-Fixable Viability Stain 510 | BioLegend | 564406 |
| APC-Cy7-anti-mouse CD45.2 | BD | 560694 |
| FITC-anti-mouse CD3e | BD | 553061 |
| PE/Cy7-anti-mouse CD4 | BD | 552775 |
| PerCP-Cy5.5-anti-mouse CD8a | BD | 551162 |
| BV421-anti-mouse CD44 | BD | 563970 |
| APC-anti-mouse CD62L | BD | 553152 |
| PE-Cy7-anti-mouse IFN-γ | BD | 557649 |

| Reagent/Resource | Reference or source | Identifier or catalogue number |
|---|---|---|
| PE-anti-mouse granzyme-B | BD | 562462 |
| BV421-anti-mouse CD107a | BioLegend | 121618 |
| APC-anti-mouse perforin | BioLegend | 154304 |
| PE-Cy7-anti-human IFN-γ | BD | 557643 |
| PE-mouse anti-human granzyme-B | BD | 562462 |
| APC-Cy7-anti-human CD45 | BioLegend | 368516 |
| FITC-anti-human CD3 | BioLegend | 300440 |
| PerCP-Cy5.5-anti-human CD8a | BioLegend | 300924 |
| PE-anti-human CD19 | BioLegend | 982402 |
| BV421-anti-human CD20 | BioLegend | 302330 |
| APC-Cy7-anti-human CD107a | BioLegend | 328630 |
| PE-Cy7-anti-human perforin | BioLegend | 353316 |
| Rig-I (D33H10) Rabbit mAb | CST | 4200S |
| Phospho-Akt (Thr308) (D25E6) XP® Rabbit mAb | CST | 13038S |
| Akt (pan) (C67E7) Rabbit mAb | CST | 4691S |
| β-Actin (13E5) Rabbit mAb | CST | 93473 |
| GAPDH monoclonal antibody | Proteintech | 60004-1-lg |
| Anti-CD3 antibody [SP162] | Abcam | ab135372 |
| CD8α (C8/144B) Mouse mAb | CST | 70306 |
| NCAM1 (CD56) (123C3) Mouse mAb | CST | 3576 |
| DDX58 Rabbit Polyclonal Antibody | HuaBio | HA500374 |
| GPC3 Mouse Monoclonal Antibody | HuaBio | EM1709-60 |
| Monoclonal Anti-Cytokeratin, pan antibody | Sigma | C2562 |
| CD3 Monoclonal Antibody (17A2) | eBioscience | 16-0032-82 |
| CD28 Monoclonal Antibody (37.51) | eBioscience | 16-0281-82 |
| InVivoMab rat IgG2a isotype control | BioXcell | BE0089 |
| InVivoMab anti-mouse CD8α | BioXcell | BE0004-1 |

| Reagent/Resource | Reference or source | Identifier or catalogue number |
|---|---|---|
| InVivoMab anti-mouse PD-1 | BioXcell | BE0146 |
| **Chemicals, Enzymes and other reagents** | | |
| DMEM high glucose | ThermoFisher | C11995500BT |
| RPMI Medium 1640 | ThermoFisher | C11875500BT |
| Foetal bovine serum (FBS) | ExCell Bio | FSP500 |
| LY294002 | Selleckchem | S1105 |
| Wortmannin | Selleckchem | S2758 |
| MK-2206 2HCl | Selleckchem | S1078 |
| GSK690693 | Selleckchem | S1113 |
| 2-deoxy-D-glucose (2-DG) | Selleckchem | S4701 |
| PMA/TPA | InvivoGen | tlrl-pma |
| Ionomycin | TargetMol | T11665 |
| GolgiStop | BD | 554724 |
| **Software** | | |
| GraphPad Prism 8.0 | https://www.graphpad.com | N/A |
| FlowJo_v10 | https://www.bdbiosciences.com/zh-cn/products/software/flowjo-v10-software | N/A |

## Methods and protocols

### *Cell lines and cell culture*

All of the cell lines were cultured with 5% $CO_2$ at 37 °C in high-glucose DMEM supplemented with 10% heat-inactivated foetal bovine serum and 100 U penicillin/streptomycin. All the cell lines were recently authenticated and confirmed to be free of mycoplasma contamination.

## Data collection

Single-cell transcriptomic data were collected from the Gene Expression Omnibus (GEO, https://www.ncbi.nlm.nih.gov/geo/) database. Four cancer datasets were used in our research: metastatic melanoma (GSE72056) (Tirosh et al, 2016), HNSCC (GSE139324) (Cillo et al, 2020), colon cancer (GSE146771) (Zhang et al, 2020), and HCC and ICC (GSE151530) (Ma et al, 2021) datasets. The metastatic melanoma, HNSCC, HCC and ICC datasets were used in this study. However, a Smart-seq2 colon cancer dataset was used in this study. Finally, a total of 86 patients (metastatic melanoma = 13, HNSCC = 26, colon cancer = 10, HCC and ICC = 37) were included in the study.

## Flow cytometry analysis

The tumours and spleens were harvested from the mice. The single-cell suspensions were incubated in the dark with fluorescent dye-labelled antibodies at 4 °C for 30 min and then washed with a buffer (1% FBS in PBS). Fixation and permeabilization were performed, and then the cells were stained with the corresponding antibodies for intracellular

staining according to the standard operating manual (BD, 554714). FlowJo (version 10) software was used to analyse the data, such as the proportion of immune cells, absolute count, cytokine secretion, and mean fluorescence intensity.

## Purification of CD8⁺ T cells and detection of intracellular cytokines

Naïve or total CD8⁺ T cells were isolated from the spleens of 6-week-old *Rig-I*⁺/⁺ and *Rig-I*⁻/⁻ mice. CD8⁺ T cells were further purified using the MojoSort™ Mouse CD8⁺ T-cell isolation kit (BioLegend, 480044 and 480035). After stimulation with PMA (100 ng/ml), ionomycin (2 µg/ml) and GolgiStop (1:1500 dilution) for 4 h in vitro, the levels of intracellular cytokines of CD8⁺ T cells purified from *Rig-I*⁺/⁺/*Rig-I*⁻/⁻ spleens were detected directly using flow cytometry. Optionally, fresh splenocytes or purified naïve CD8⁺ T cells were cultured in 96-well U-bottomed plates in a complete RPMI medium supplemented with plate-bound anti-CD3 (0–5 µg/ml) and soluble anti-CD28 (1.25 µg/ml). Small molecule inhibitors were incubated with CD8⁺ T cells in vitro for the indicated times.

## Western blot (WB) analysis

Cell lysates were prepared using RIPA buffer (Beyotime, P0013B) containing a protease inhibitor mixture (Beyotime, P1005). As previously mentioned, Western blotting of the target protein was performed using the corresponding primary antibody, followed by exposure to an HRP-coupled secondary antibody. The reaction bands were observed using an ECL reagent (Millipore, WBKLS0500), and the relative intensities of the protein bands were analysed using ImageJ software (NIH).

## Molecular constructs and knockout of human RIG-I

The CRISPR/Cas9 genome-editing system was used to knock out human RIG-I. Three individual sgRNAs targeting RIG-I were cloned and inserted into the pLentiCRISPR v2 vector. The primers synthesised by Beijing Ruibiotech Co., Ltd. were shown in Appendix Table S1. The RIG-I knockout lentivirus was collected from the supernatant of HEK293T cells after cotransfection with the three plasmids (psPAX2: pMD2.G: pLentiCRISPR v2 = 5: 3: 2) for 72 h. After 3 h of centrifugal infection with polybrene (8 µg/ml), the medium was replaced with a fresh complete medium. The knockout efficiency was detected by Western blot, and the most efficient sgRNA was selected for subsequent experiments.

## Multiplex IHC

The levels of cell subsets in the TME were evaluated using multiplex IHC. A PANO 7-plex IHC kit (Panovue, 0004100100) was used for multiple immunofluorescence staining, and different primary antibodies were sequentially used. Then, horseradish peroxidase-labelled secondary antibody and tyramide signal amplification (TSA) solution were added and incubated. After each incubation with the TSA solution, the sections were subjected to microwave heat treatment. After all the antigens were labelled, the nuclei were stained with 4′-6′-diamino-2-phenylindole (DAPI, Sigma–Aldrich). Olympus VS200 MTL (Olympus Germany) and Olympus UPLXAPO20X objective lenses were used to obtain whole-slide multicolour fluorescence images, and QuPath software was used to perform the quantitative pathological analysis of the whole-slide multicolour fluorescence images.

## Generation of LCLs and EBV-specific CTLs

Firstly, $2–5 \times 10^6$ PBMCs isolated from a healthy donor (male, 35-year-old) in a 12-well plate were resuspended in 500 µl RPMI containing 100–500 infectious units of EBV. Half the volume of media was changed every 4 days, and LCL colony formation was evident under the microscope at 3-4 weeks postinfection. Then, the cells were transferred to a T25 flask, or appropriate flasks continued to be cultured and cryopreserved. To generate cytotoxic T lymphocytes (CTLs) that primarily recognise the immunodominant underlying antigen of EBV, PBMCs were prepared and stored in liquid nitrogen prior to use. The feeder cells were prepared by collecting $1–2 \times 10^6$ autologous LCLs and irradiating them with a single dose of 80 Gy. Then, the resuspended PBMCs were cultured with feeder cells at a final ratio of 30:1 in a medium supplemented with 120 IU/mL IL-2 (R&D, 202-IL-050). The medium containing IL-2 was changed every 3 to 4 days until the end of the culture period. On day 21, the number of viable cells was counted using Trypan blue exclusion. The amplified CTLs were stored for further experiments.

## Animal study

C57BL/6J WT or B-NDG mice were purchased from Zhuhai BesTest Bio-Tech. (Guangdong, China). C57BL/6J *Rig-I*⁻/⁻ mice were provided by Pro. Jiang Zhu (Shanghai Institute of Haematology). All mice were housed under specific pathogen-free conditions with light/dark cycles and were used at ages between 6 to 14 weeks. All mice of the same sex were used in the same experiment and were randomly assigned to each treatment group. For primary tumorigenesis experiments, MC38 ($2 \times 10^5$ cells), B16F10 ($1 \times 10^5$ cells), Hepa1-6 ($3 \times 10^6$ cells), MC38-OVA ($1 \times 10^6$ cells) or LCL ($1 \times 10^7$ cells) cells were injected subcutaneously into randomly selected in mice a final volume of 100 µl. Tumour growth was measured using digital callipers every 2–4 days. At the indicated time points, the tumours were excised and subjected to flow cytometry analysis.

Some groups of mice received 100 µg/mouse anti-mouse CD8α to deplete CD8⁺ T-cell subsets. Anti-mouse CD8α or anti-rat IgG2a isotype control was injected i.p. at 100 µg/mouse as described in the figure legends. For immunotherapy, some groups of mice received 250 µg/mouse control immunoglobulin (cIg) or anti–PD-1 mAb as indicated in the figure legends.

## Statistics

No blinding was performed in all experiments of this study. For in vitro experiments, at least three biological replicates were performed, according to good laboratory practices. For in vivo or ex vivo experiments a priori error probabilities were set to $\alpha = 0.05$ and $\beta = 0.20$ to calculate the number of animals needed. No sample/animal was excluded from the study. The data were presented as the means ± SEMs. Significant differences were evaluated using two-tailed unpaired Student's *t*-test, the two-tailed Mann–Whitney *U*-test, one-way ANOVA, two-way ANOVA or the log-rank test.

**The paper explained**

**Problem**

Immunotherapy with immune checkpoint inhibitors has brought hope to cancer treatment, but the low response rate limits their widespread use. Therefore, studies exploring and developing novel effective targets for immunotherapy are urgently needed, and these targets will be of significant clinical importance in improving patient treatment response rates and extending survival.

**Results**

We analyzed tumour-infiltrating lymphocytes (TILs) using bioinformatics and found that RIG-I is specifically upregulated in CD8$^+$ T cells, especially in terminally exhausted CD8$^+$ T cells. Ex vivo experiments confirmed that RIG-I knockout enhances the development of CD8$^+$ T cells and the secretion of anti-tumour cytokines. In mouse models with *Rig-I* knockout, the growth of various tumours was suppressed. Moreover, the adoptive transfer of RIG-I-knockout CD8$^+$ T cells significantly inhibited tumour growth in mice, increased the infiltration of CD8$^+$ T cells, and enhanced the secretion of anti-tumour cytokines. Mechanistically, the activation of CD8$^+$ T cells by TCR signalling leads to the upregulation of RIG-I, which in turn inhibits the AKT/glycolysis signalling pathway, negatively regulating anti-tumour function. For tumours insensitive to PD-1 antibodies, the combination of RIG-I knockout with PD-1 antibodies shows promising therapeutic effects.

**Impact**

In this study, we identified a new intracellular checkpoint, RIG-I, which resists the anti-tumour activity of CD8$^+$ T cells in various solid tumours through the AKT/glycolysis signalling pathway. Furthermore, targeting RIG-I in CD8$^+$ T cells can slow tumour growth. As a novel intracellular checkpoint in CD8$^+$ T cells within the tumour microenvironment, RIG-I represents a promising therapeutic target that can enhance the function of CD8$^+$ T cells, particularly in the treatment of solid tumours that are resistant to PD-1 antibodies.

All the statistical analyses were performed using GraphPad Prism software (version 8.0). The following indicators and values were used to indicate statistical significance: $*P < 0.05$, $**P < 0.01$, $***P < 0.001$ and $****P < 0.0001$. The exact $p$ values are shown in each panel of the figures.

## Human and animal ethics

Informed consent was obtained from each healthy donor and patient prior to participation. The collection of peripheral blood samples from healthy donors and the collection of paraffin-embedded tissue from patients with HCC and colon cancer were approved by the Clinical Research Ethics Committee of Zhuhai People's Hospital (No. 2022089). The experiments conformed to the principles established in the WMA Declaration of Helsinki and the Department of Health Services Belmont Report. All animal procedures were approved by the Experimental Animal Ethics Committee of Zhuhai People's Hospital (No. 2022082608).

## Data availability

The datasets produced in this study are available in the following databases: RNA-seq data: Gene Expression Omnibus GSE230045 (https://www.ncbi.nlm.nih.gov/geo/query/acc.cgi?acc=GSE230045).

RNA-seq data: Genome Sequence Archive CRA010683 (https://ngdc.cncb.ac.cn/gsa/browse/CRA010683). The FAC source data: BioStudies S-BSST1481. The multiplex IHC source data: BioStudies S-BSST1471.

The source data of this paper are collected in the following database record: biostudies: S-SCDT-10_1038-S44321-024-00136-9.

## Peer review information

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

## Acknowledgements

We would like to express our gratitude to the healthy individuals and patients who donated samples, and to Prof. Dong-Ming Kuang for his guidance on this project. We thank Prof. Jiang Zhu (Shanghai Institute of Haematology) for providing the *Rig-I* knockout mice. This study was supported by research grants from the National Natural Science Foundation of China (Nos. 82230067, 82103433, 32470961, 82204415, 32200936, 82103306 and 82304589), the Guangdong Provincial Key Laboratory of Tumour Interventional Diagnosis and Treatment (No. 2021B1212040004), the Guangdong Basic and Applied Basic Research Foundation (Nos. 2024A1515012563, 2020A1515110013, 2021A1515220187 and 2021A1515110561), the Shenzhen Science and Technology Innovation Commission projects (No. JCYJ20220818101808018), the Science and Technology Planning Project of Guangzhou (No. SL2024A03J00598) and the Scientific Research start-up Project of Zhuhai People's Hospital (No. 2021KYQD-04).

## Author contributions

**Xiaobing Duan**: Conceptualisation; Data curation; Formal analysis; Funding acquisition; Investigation; Methodology; Writing—original draft; Project administration; Writing—review and editing. **Jiali Hu**: Data curation; Formal analysis; Validation; Investigation; Methodology. **Yuncong Zhang**: Data curation; Formal analysis; Investigation; Methodology. **Xiaoguang Zhao**: Resources; Software; Formal analysis; Supervision; Validation; Investigation; Methodology. **Mingqi Yang**: Resources; Software; Formal analysis. **Taoping Sun**: Resources; Data curation; Software. **Siya Liu**: Resources; Data curation; Formal analysis; Validation. **Xin Chen**: Resources. **Juan Feng**: Resources. **Wenting Li**: Resources; Data curation; Software; Formal analysis; Validation. **Ze Yang**: Resources; Data curation; Software; Formal analysis; Validation; Methodology. **Yitian Zhang**: Resources; Data curation; Software; Supervision; Validation; Methodology. **Xiaowen Lin**: Resources; Data curation; Software; Supervision; Validation; Methodology. **Dingjie Liu**: Data curation; Software; Formal analysis; Supervision; Validation; Methodology. **Ya Meng**: Resources; Data curation; Formal analysis; Methodology. **Guang Yang**: Resources; Data curation; Formal analysis; Methodology. **Qiuping Lin**: Resources; Supervision; Validation; Investigation; Methodology. **Guihai Zhang**: Resources; Software; Formal analysis; Supervision; Validation; Investigation; Methodology. **Haihong Lei**: Methodology. **Zhengsheng Yi**: Data curation; Formal analysis; Methodology. **Yanyan Liu**: Resources; Software. **Xiaobing Liang**: Resources; Software; Validation. **Yujuan Wu**: Resources; Software. **Wenqing Diao**: Resources; Software. **Zesong Li**: Resources; Software. **Haihai Liang**: Resources; Supervision; Validation. **Meixiao Zhan**: Supervision; Validation; Investigation; Visualisation; Methodology; Writing—original draft. **Hong-Wei Sun**: Conceptualisation; Resources; Formal analysis; Supervision; Validation; Investigation; Visualisation; Methodology; Writing—original draft; Project administration; Writing—review and editing. **Xian-Yang Li**: Conceptualisation; Resources; Formal analysis; Supervision; Funding acquisition; Validation; Investigation; Methodology; Writing—original draft; Project administration; Writing—review and editing. **Ligong Lu**: Conceptualisation; Resources; Funding acquisition; Project administration; Writing—review and editing.

Source data underlying figure panels in this paper may have individual authorship assigned. Where available, figure panel/source data authorship is listed in the following database record: biostudies:S-SCDT-10_1038-S44321-024-00136-9.

## Disclosure and competing interests statement

XBL is an employee of Huixin Life Science. The authors declare no competing interests.

# Expanded View Figures

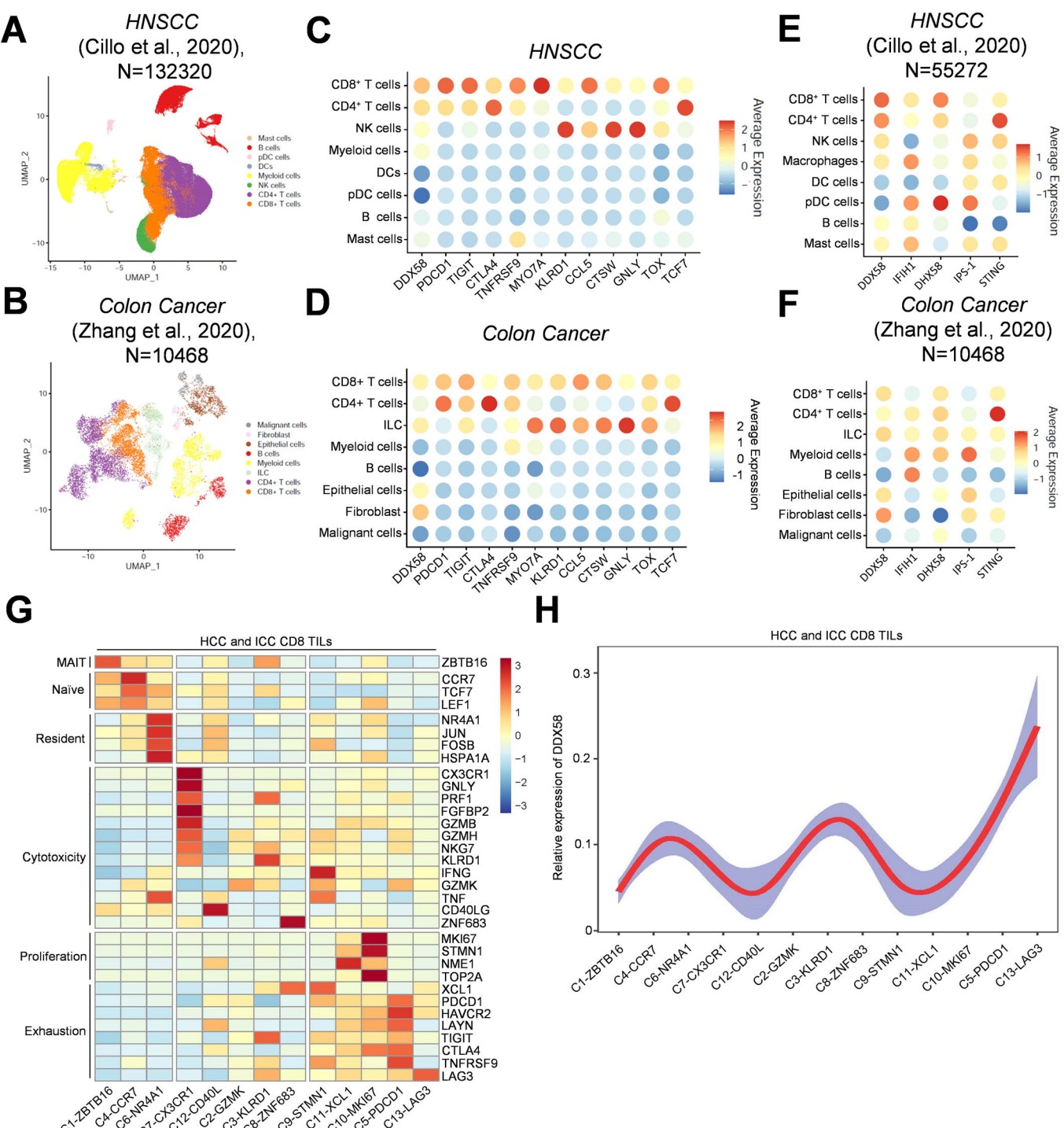

**Figure EV1.  Screening single-cell sequencing data and verification of RIG-I upregulation in CD8+ T cells infiltrating the TME.**

(A, B) Cluster analysis of cell populations in HNSCC and colon cancer, and expression levels of the *DDX58* gene in each subpopulation. (C, D) Relative expression levels of *DDX58, PDCD1, TIGIT, CTLA-4* and *TOX*-associated exhaustion genes in each subpopulation in HNSCC (C) and colon cancer (D). (E, F) Relative expression levels of *DDX58, IFIH1, DHX58, IPS-1* and *STING* in HNSCC (E) and colon cancer (F) tissues. (G) Heatmap indicating the expression of selected gene sets in CD8+ T-cell subtypes infiltrating HCC and ICC. (H) Line chart showing the relative expression patterns of DDX58 in each CD8+ T-cell subtype. Source data are available online for this figure.

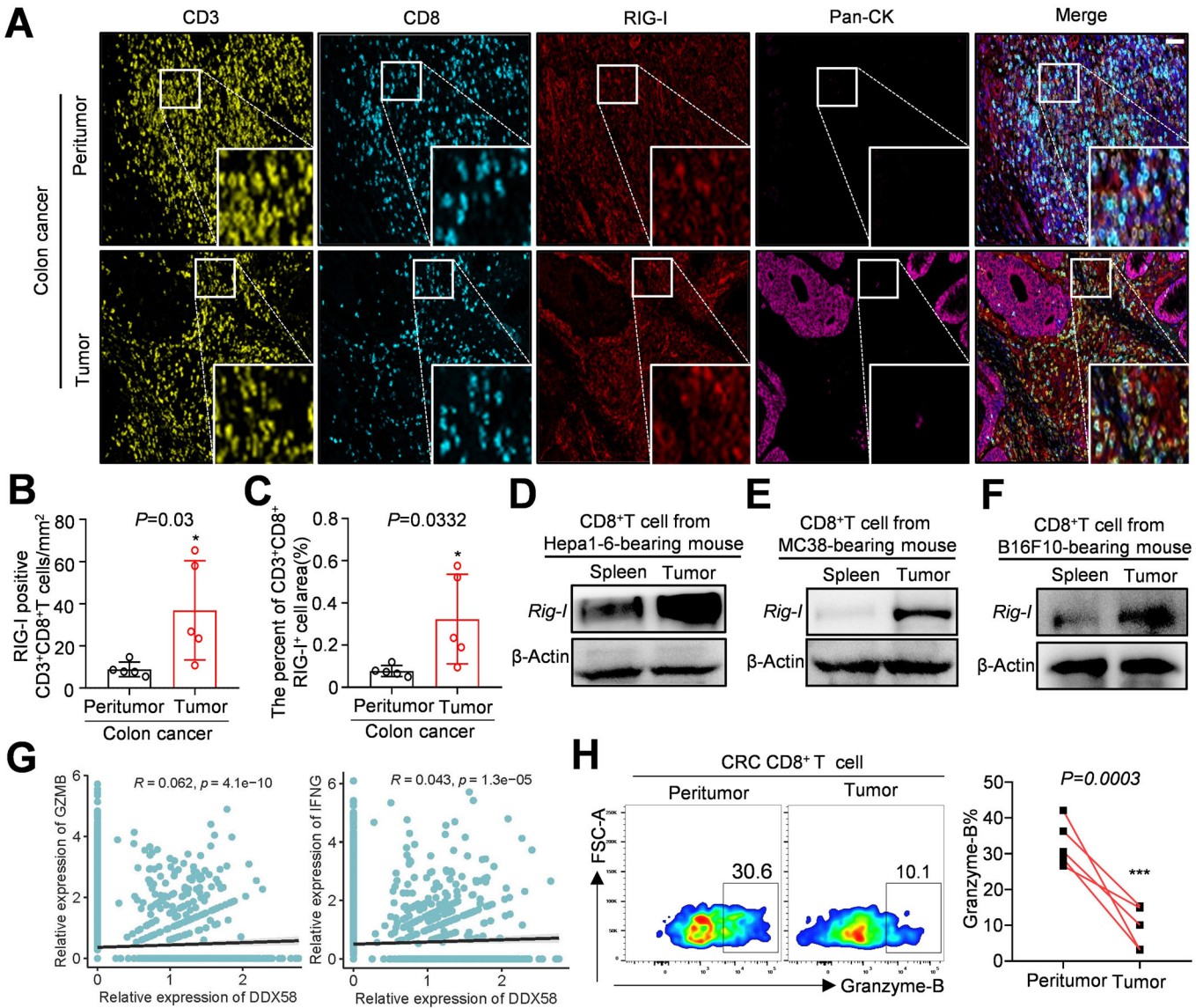

**Figure EV2. The upregulation of RIG-I expression was validated in CD8+ T cells infiltrating the TME.**

(A–C) Multiple immunohistochemistry and statistical analysis of human colon cancer specimens. White scale bar = 50 μm. (D–F) The expression of *Rig-I* in CD8+ T cells from the spleens and tumours of different tumour-bearing mice was detected using Western blotting. (G) Correlation analysis of DDX58 expression with GZMB and IFNG expression. (H) The expression of granzyme-B secreted by infiltrating CD8+ T cells in the peritumor and tumour tissues of CRC by flow cytometry after stimulating with PMA/ionomycin and GolgiStop. Data information: The data represented different numbers (*n* = 5) of biological replicates and were shown as the means ± SEMs. Two-tailed unpaired Student's test was used in (B) (*P* = 0.03), (C) (*P* = 0.0332). The Pearson correlation test was used in (G) (Exact *p* values were reported on graphs). The values of *P* and *R* are shown in the figure. Two-tailed paired Student's test was used in (H) (*P* = 0.0003). *$P < 0.05$, ***$P < 0.001$, compared with the peritumoral group. Source data are available online for this figure.

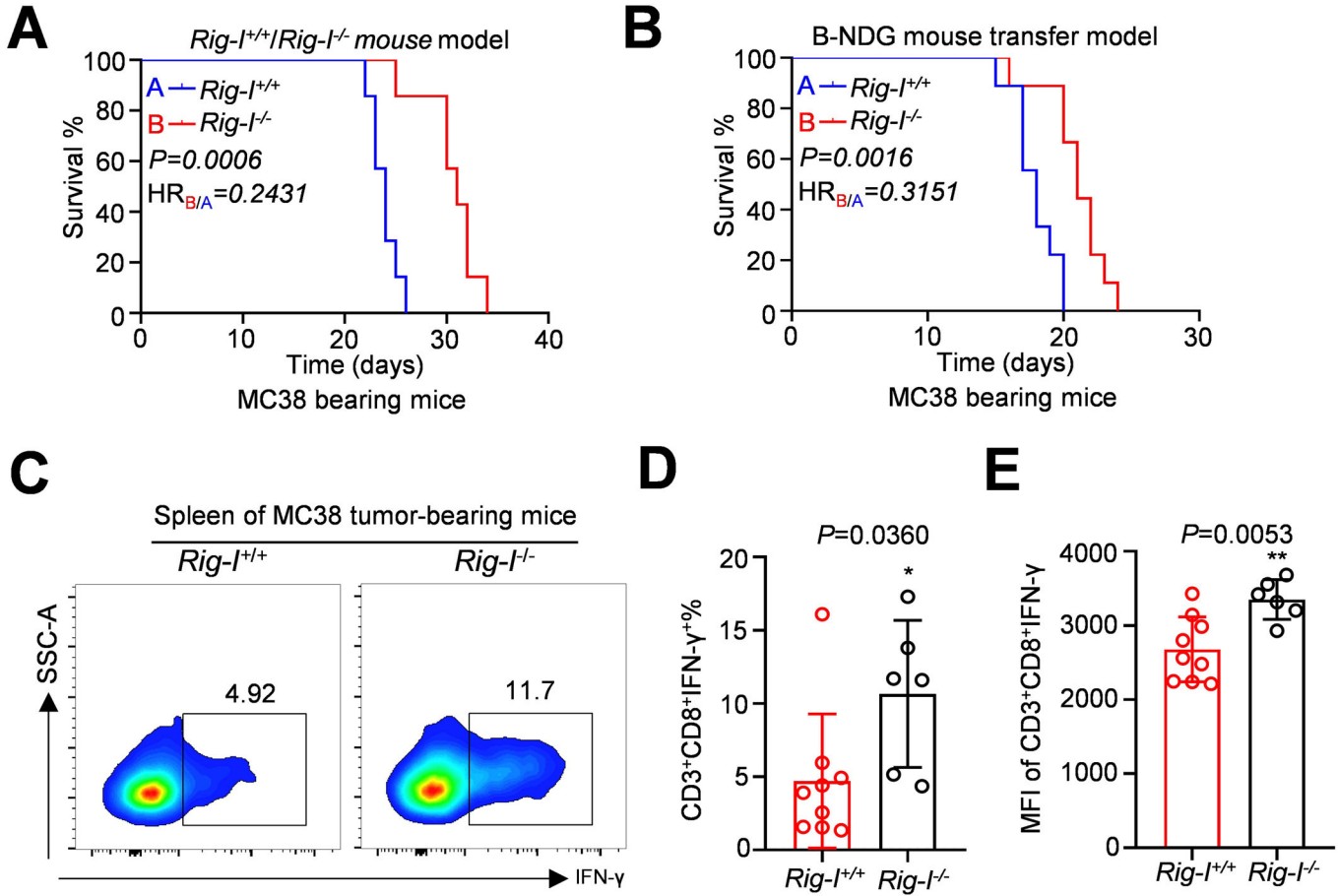

**Figure EV3. Rig-I knockout enhanced the anti-tumour function.**

(A) Survival curves of MC38 tumour-bearing *Rig-I^{+/+}* and *Rig-I^{−/−}* mice. (B) Survival curves of MC38 tumour-bearing B-NDG mice after the adoptive transfer of *Rig-I^{+/+}* or *Rig-I^{−/−}* CD8^+ T cells. (C–E) Flow cytometry was used to detect the proportion (C, D) and mean fluorescence intensity (E) of CD8^+ T cells that produced IFN-γ in the spleens of tumour-bearing *Rig-I^{+/+}* and *Rig-I^{−/−}* mice after stimulation with PMA/ionomycin and GolgiStop. Data information: The data represent different numbers ($n = 7$ for (A) and $n = 9$ for (B) of biological replicates. The data represent different numbers ($n = 6$ or 9 for D and E) of biological replicates and were shown as the means ± SEMs. The log-rank test was used in (A) ($P = 0.0006$) and (B) ($P = 0.0016$). A two-tailed Mann–Whitney $U$-test was used in (D) ($P = 0.0360$). Two-tailed unpaired Student's test was used in (E) ($P = 0.0053$). *$P < 0.05$, **$P < 0.01$ and ***$P < 0.001$ compared with the *Rig-I^{+/+}* group. Source data are available online for this figure.

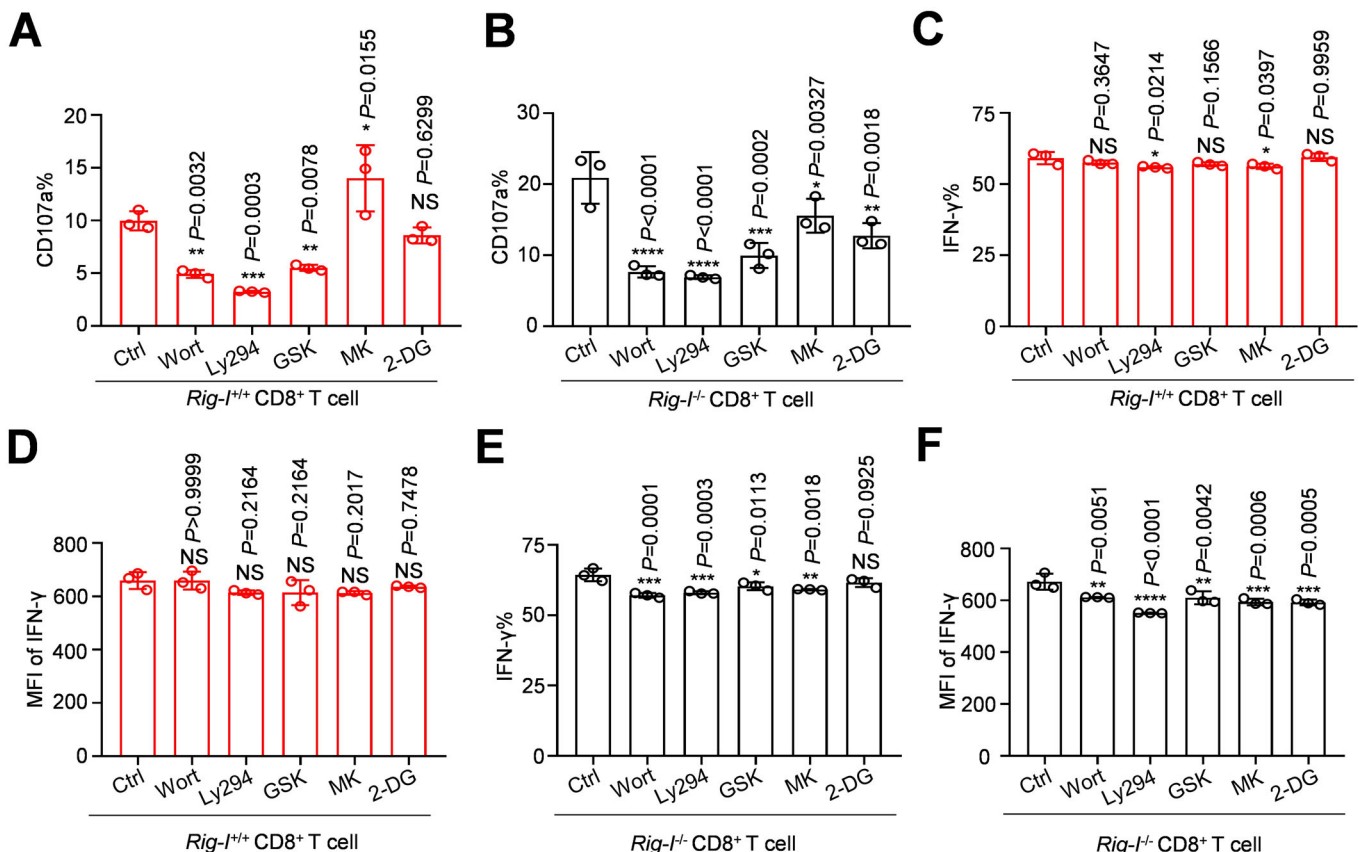

**Figure EV4.** ***Rig-I* inhibited the PI3K/AKT/glycolysis signalling pathway to protect against the anti-tumour effects of CD8$^+$ T cells.**

(A–F). Treatment of *Rig-I$^{+/+}$* and *Rig-I$^{-/-}$* CD8$^+$ T cells with α-CD3/α-CD28 followed by stimulation with inhibitors of PI3K, AKT and glycolysis. The proportions or mean fluorescence intensities of CD107a (A, B) and IFN-γ (C–F) were detected using flow cytometry after stimulation with PMA/ionomycin and GolgiStop. Data information: The data represented different numbers ($n = 3$) of biological replicates and were shown as the means ± SEMs. One-way ANOVA with Tukey's correction was used for multiple comparisons in (A–F) (Exact $p$ values were reported on graphs). *$P < 0.05$, **$P < 0.01$, ***$P < 0.001$, ****$P < 0.001$, and NS not significant compared with the Ctrl group. figure. Ctrl control, Wort wortmannin, Ly294 Ly294002, GSK GSK690693, 2-DG 2-deoxy-ᴅ-glucose. Source data are available online for this figure.

