## [Peer Review File · EMBO Molecular Medicine]

RIG-I is an intracellular checkpoint that limits CD8+ T-cell antitumour immunity

Xiaobing Duan, Jiali Hu, Yuncong Zhang, Xiaoguang Zhao, Mingqi Yang, Taoping Sun, Siya Liu, Xin Chen, Juan Feng, Wenting Li, Ze Yang, Yitian Zhang, Xiaowen Lin, Dingjie Liu, Ya Meng, Guang Yang, Qiuping Lin, Guihai Zhang, Haihong Lei, Zhengsheng Yi, Yanyan Liu, Xiaobing Liang, Yujuan Wu, Wenqing Diao, Zesong Li, Haihai Liang, Meixiao Zhan, Hong-Wei Sun, Xian-Yang Li, and Ligong Lu

Corresponding authors: Ligong Lu (luligong1969@jnu.edu.cn) , Hong-Wei Sun (shongw@mail3.sysu.edu.cn), Xiaobing Duan (duanxb3@mail3.sysu.edu.cn), Xian-Yang Li (lixianyang@ext.jnu.edu.cn)

Review Timeline:

Submission Date:	22nd Nov 23
Editorial Decision:	18th Dec 23
Revision Received:	6th Jul 24
Editorial Decision:	31st Jul 24
Revision Received:	27th Aug 24
Accepted:	29th Aug 24

Editor: Poonam Bheda

Transaction Report:

18th Dec 2023

Dear Prof. Lu,

Thank you for the submission of your manuscript to EMBO Molecular Medicine. We have now received feedback from the three reviewers who agreed to evaluate your manuscript. As you will see from the reports below, the referees acknowledge the interest of the study; however, they also express concerns on some of the main conclusions and whether they are sufficiently supported by the data. Editorially we have discussed these issues also with the reviewers in a cross-commenting session and agree that the main issues are as follows:

- in line with comments from both Reviewers 1 and 3, the claims on a RIG-I intrinsic role in CD8+ T cells needs further substantiation, as RIG-I is also likely expressed in innate immune cells and the use of a whole KO mouse is a limitation. Editorially we would find a combination of some additional experimental analyses (such as stainings) as well as toning down the conclusions to be sufficient to address these issues.
- The discrepancies pointed out by Reviewer 3 and agreed upon during cross-commenting on RIG-I expression will need to be addressed.

Addressing the remaining reviewers' concerns in full in a point-by-point response will be necessary for further considering the manuscript in our journal. Please note that we have also discussed Reviewer 3's concern that the RIG-I KO mice are being compared with different WT mice - while we agree that comparing littermates would be ideal, as they are both from the same C57BL/6J background and we understand that it is not always possible when mutant mice are provided from a different source than the WT, this concern does not need to be addressed experimentally.

Please note that EMBO Molecular Medicine encourages a single round of revision only, which will be followed by a re-review, and therefore, acceptance or rejection of the manuscript will depend on the completeness of your responses included in the next, final version of the manuscript. For this reason, and to save you from any frustrations in the end, I would strongly advise against returning an incomplete revision. If you would like to discuss further the points raised by the referees, I am available to do so via email or video. Let me know if you are interested in this option.

We are expecting your revised manuscript within three months, if you anticipate any delay, please contact us.

We require:

- 1) A .docx formatted version of the manuscript text (including legends for main figures, EV figures and tables). Please make sure that the changes are highlighted to be clearly visible.
- 2) Individual production quality figure files as .eps, .tif, .jpg (one file per figure). For guidance, download the 'Figure Guide PDF' (<https://www.embopress.org/page/journal/17574684/authorguide#figureformat>).
- 3) At EMBO Press we ask authors to provide source data for the main figures. Our source data coordinator will contact you to discuss which figure panels we would need source data for and will also provide you with helpful tips on how to upload and organize the files.
- 4) A .docx formatted letter INCLUDING the reviewers' reports and your detailed point-by-point responses to their comments. As part of the EMBO Press transparent editorial process, the point-by-point response is part of the Review Process File (RPF), which will be published alongside your paper.
- 5) A complete author checklist, which you can download from our author guidelines (<https://www.embopress.org/page/journal/17574684/authorguide#submissionofrevisions>). Please insert information in the checklist that is also reflected in the manuscript. The completed author checklist will also be part of the RPF.
- 6) Please note that all corresponding authors are required to supply an ORCID ID for their name upon submission of a revised manuscript.
- 7) It is mandatory to include a 'Data Availability' section after the Materials and Methods. Before submitting your revision, primary datasets produced in this study need to be deposited in an appropriate public database, and the accession numbers and database listed under 'Data Availability'. Please remember to provide a reviewer password if the datasets are not yet public (see <https://www.embopress.org/page/journal/17574684/authorguide#dataavailability>).

This study includes no data deposited in external repositories.

8) For data quantification: please specify the name of the statistical test used to generate error bars and P values, the number (n) of independent experiments (specify technical or biological replicates) underlying each data point and the test used to calculate p-values in each figure legend. The figure legends should contain a basic description of n, P and the test applied. Graphs must include a description of the bars and the error bars (s.d., s.e.m.). Please provide exact p values.

13) Author contributions: CRedit has replaced the traditional author contributions section because it offers a systematic machine readable author contributions format that allows for more effective research assessment. Please remove the Authors Contributions from the manuscript and use the free text boxes beneath each contributing author's name in our system to add specific details on the author's contribution. More information is available in our guide to authors.

Share synopsis text and image, as well as eTOC:

Please note that these would be the final versions and changes during proofing are usually not allowed

16) As part of the EMBO Publications transparent editorial process initiative (see our Editorial at <http://embomolmed.embopress.org/content/2/9/329>), EMBO Molecular Medicine will publish online a Review Process File (RPF)

to accompany accepted manuscripts.

In the event of acceptance, this file will be published in conjunction with your paper and will include the anonymous referee reports, your point-by-point response and all pertinent correspondence relating to the manuscript. Let us know whether you agree with the publication of the RPF and as here, if you want to remove or not any figures from it prior to publication. Please note that the Authors checklist will be published at the end of the RPF.

I look forward to receiving your revised manuscript.

Yours sincerely,

Poonam Bheda

Poonam Bheda, PhD
Scientific Editor
EMBO Molecular Medicine

**** Reviewer's comments ****

Referee #1 (Remarks for Author):

Reviewer's Report for: RIG I is an intracellular checkpoint that limits CD8 T cell antitumor immunity

This manuscript focuses on demonstrating a new role for the retinoic acid-inducible I (RIG-I) in the inhibition of CD8⁺ T cell function. The authors hypothesize that RIG-I functions as an intracellular checkpoint that limits anti-tumour CD8⁺ T cell immunity. They show that RIG-I expression is enhanced in CD8 TILs in single cell datasets from different human cancers. They use a RIG-I KO mouse model to assess the effects of RIG-I on splenic CD8 T cells phenotype using RNA sequencing and flow cytometry. Using these same mice, the authors show that tumours develop slower when RIG-I is not present (KO vs. WT) and that the delayed growth requires CD8 T cells to be present also. They show that the CD8 T cells in tumour-bearing RIG-I KO mice are increased in number and production of effector cytokines, including antigen-specific CD8 T cells in the tumour. They show by western blot that RIG-I KO enhances pAKT and enhances CD8 T cell activation, which is targetable with small molecule inhibitors of RIG-I. Finally, they demonstrate delayed tumour growth in two tumour model and synergy with PD-1 blockade in PD-1 resistant but not susceptible models. Overall, the data presented in the manuscript is novel, interesting to the field and well produced with appropriate controls and experimental rigor to answer experimental questions. Although, I feel the writing could be improved with some editing to achieve greater clarity and brevity in the text.

I have included below my comments for the authors consideration:

- 102 - In the abstract, please clarify, how RIG-I activation is induced by 'overactivated T cells'?
- The introduction has an appropriate level of detail and background.
- Overall, the results sections are too lengthy, with much explanation and extrapolation, which should be restricted to the introduction or discussion. Introductory sentences in results should be kept as brief as necessary/possible to set the scene appropriately.
- 152 - RIG-I is expected to be expressed in innate immune cells in addition to its observed upregulation in CD8 T cells. Did the authors examine/exclude expression in innate cells e.g. dendritic cells in tumours, which (in mice at least), also express CD8, in their analyses? The authors could look specifically at cytotoxicity using ex vivo CTL assays of RIG-I Ko and WT CD8⁺ T cells.
- Figure 3 - The weakness of using the RIG-I KO mouse in experiments measuring tumour control, and looking specifically at CD8⁺ T cells, is that the effect of RIG-I KO on other immune cells, including innate cells is missed and the effect of RIG-I KO in CD8 T cells specifically may be over-interpreted. It is noted that differences are often subtle in these tumour models. Similarly, experiments depleting CD8 T cells almost universally result in increased tumour progression/growth, so require careful interpretation.
- 198 - OVA antigen can induce de novo immune responses in mice, however, this will represent only a portion of the tumour specific repertoire. Thus, the authors may be missing important information if only considering responses to this model antigen.

- 222 - 'knocking out RIG-I in CD8+ T cells enhances their antitumour function in vivo mainly by increasing the proportion of CD8+ TIL and secretion of IFN γ ' - Function is not related to proportion of cells, and IFN γ is a simple marker of CD8 function, could this be quantified in a beyond one marker in flow cytometry?
- 229 - "T cell exhaustion is a type of cell population" - rather it is a phenotype or state of T cells.
- 238 - Overly verbose, stick to just providing results in the results section.
- 250 - Clarity and brevity is preferred over broad statements, particularly indicate the data the relevant data indicating the exhaustion state as opposed to looking at effector cytokine release alone.
- 256 - This is perhaps information that should be moved to the introduction, not presented in the results section.
- 278 - 'overactivated T cell can induce RIG-I upregulation in CD8 T cells' - this statements is confusing and should be explained, what induced RIG-I upregulation and where in the sequence of events in CD8 suppression does this occur?
- 286 - Do they authors mean targeting RIG-I or KO of RIG-I, enhancing CD8 T cells?
- 296 - "breeding" layers - do the authors mean feeders?
- 306 - What is meant by" ability to weaken human tumors in vivo". Please be as specific as possible in conclusions and interpretation of results.
- 322 - The finding that RIG-I KO doesn't improve PD-1 efficacy in susceptible tumour models is interesting, while in resistant models it does seem to. Could the authors comment on this further regarding mechanisms of t cell suppression that differ between the two models?
- 1405 words for the discussion is too long in my opinion, but this may be within the journal guidelines. It could be improved with more clarity and brevity, avoiding repetition of comments and results presented already within the manuscript. At some points an unscientific writing style that is not acceptable for a peer reviewed publication is used (430). I feel the discussion needs improvement to be at the level necessary for publication.
- Some methods (505, 532, 552/3/7) need more detail, 561 requires less detail (cells were removed from the incubator). 577 - provide info on doses given and schedule.
- 588 - animal ethics approvals MUST BE stated.
- Figure 1 - Why not show consistent markers across HCC and CRC tumours?
- Figure 4 - Axes labels should indicate the appropriate population, else how do B/C differ? Is it CD8+CD3+ and CD8-CD3+? The axes are blurry and difficult to read.
- Figure 6 - And in general for tumour experiments, no data on survival is provided and experiments are often not taken to a survival endpoint. How effective then is RIG-I KO at improving CD8+ T cell function leading to real survival benefit?
- This is particularly relevant in the PD-1 experiments in Figure 7 in both models where survival looks achievable.

Referee #2 (Remarks for Author):

While immune checkpoint inhibitors have led to major changes in cancer treatment, many cancers do not respond to this therapeutic venue. Identifying druggable T cell targets is of high relevance to cancer immunotherapy. In the present study, the authors have identified RIG-I as an intracellular checkpoint that negatively regulates CD8 T cell activation in a cancer setting. The results are overall compelling and supportive. Specific comments follow.

1) The authors have fairly acknowledged the very recent work of Jiang et al. published in the JCI this year. Because of the high originality of the reported findings, the observation that similar results are obtained here is a strength of the study, and this should stimulate further research in the field. However, it remains important to test if Stat5 is involved in the context of the present work. This should be at the very least discussed.

2) The resolution of the figures is insufficient. This is problematic to evaluate the data in Figure 1G for instance. This should be rectified.

3) These results are reminiscent of other studies published in the field demonstrating various functions for cytosolic sensors in T cells. The authors may consider discussing this aspect more, for instance the study of Li et al., Science Translational Medicine, 2020.

4) Some typos should be corrected. For instance, in Figure 8 (dysfunctional).

In summary, this is a sound study that relies on both mouse and human data to ascribe a novel function to RIG-I. The results are relevant for cancer immunotherapy.

Referee #3 (Comments on Novelty/Model System for Author):

Comparing full RIG-I KO with WT mice from other origin may be misleading with important differences in microbiome that could explain the differences in CD8 T cell differentiation and reactivity. I would suggest to analyse littermates, perform mixed BM chimeras to analyse RIG-I competent and deficient in similar environment. To address the T cell intrinsic role of RIG-I, T cell deficient using CD4cre and RIG-I^{fl/fl} or at least adoptive transfer experiments are required.

Referee #3 (Remarks for Author):

In this study the authors wanted to investigate the impact of RIG-I expression in CD8 T cells antitumor functions. They found in public data base that this RNA sensor is overexpressed by TILs and they made the hypothesis that RIG-I could participate in the exhaustion process and tumor escape. The analysis is mainly based on the comparison between full RIG-I KO and WT mice possibly from other origin. The intrinsic role of RIG-I cannot be inferred from the experiments presented in this manuscript. Many discrepancies are present throughout the manuscript leading to potential false conclusions from the authors. The presentation of the data is suboptimal and the figure legend is not always accurate. While the role of RIG-I in CD8 T cells is interesting the experimental models used and the analysis conducted are not sufficient .

Figure 1:

The authors should consider analysing RIG-I expression by specific CD8 TILs subsets. Is RIG-I expressed by all TILs or only T_{ex} subsets, which subsets T_{pex} or T_{term} T_{ex}?

The link between RIG-I expression and the low GZMB or IFN γ expression by TILs is not obvious?

Figure2:

The RNAseq analysis is interesting but not very well detailed in the manuscript. Before focusing on key genes and pathways, the authors should first show the raw differential gene expression between RIG Wt and Ko (volcano plot showing the adj p value and FC) for example. Why did the authors selected these genes in particular in Fig 2B? Are they handpicked, higher FC or p value or related to a pathway enriched in Fig2A?

It is not clear which markers and gates were used to quantify the memory and effector cells in Fig 2K-L or in Figure S3.

Especially, with regards to the CD44-CD62L- pop present in the representative FACS plots in Fig2I?

The authors found a decrease in naïve in FigS3A but NS in Fig2J.

The authors state "To rule out the possibility that RIG-I indirectly regulates the antitumor function of CD8+ T cells through other immune cells, we purified CD8+ T cells from RIG-I^{+/+} and

RIG-I^{-/-} mouse spleens in vitro (Fig. S4). » While it rules out a bystander cell role during the PMA/ionomycin stimulation, it doesn't prove a T cell intrinsic role of RIG-I in CD8 T lymphocytes in the regulation of T cells.

The authors should prove that RIG-I is expressed by T cells in the spleen of WT mice. In Fig S1 they show RIG one is absent which suggest that the differences observed in the spleen are due to RIG-I expression by other cells. In FIG1 and S1 the authors suggest RIG-I is specific for TILs. Thus, the difference in cytokine production upon PMA/ionomycin stimulation might just reflect the higher memory frequency in RIG-I^{-/-} mice.

For the Fig S4 the authors state "Analysis of the IFN- γ , granzyme-B and the development (including naive, memory and effector cells) of CD8+ T or CD8-T cells purified from RIG-I^{+/+}/RIG-I^{-/-} spleen after stimulation with PMA, ionomycin and Golgi blockers in vitro by flow cytometry." but the figure doesn't show purified cells and no analysis of CD8- cells??? This figure looks similar to the figure S2 in terms of settings and gating.

Comparing full RIG-I KO with WT mice from other origin may be misleading with important differences in microbiome that could explain the differences in CD8 T cell differentiation and reactivity. I would suggest to analyse littermates or mixed BM chimeras to analyse RIG-I competent and deficient in similar environment.

Figure 3:

The authors show in well characterized CD8 T cells dependent models that RIG-I^{-/-} mice have lower tumor growth and conclude "Collectively, these findings underscore the direct impact of RIG-I in limiting the antitumor efficacy of CD8+ T cells." This is an overstatement given the authors already showed developmental differences in steady state conditions in RIG-I KO vs WT (Fig2). In addition, the CD8 T cell dependence of the model doesn't imply an intrinsic effect of RIG-I in CD8 T cells.

Figure 4:

The authors should consider showing cell numbers in addition to cell frequencies. They show no differences in CD8+% in figure 4A (31,9 vs 34,2%) and yet in fig 4B CD8+ T cells are shown increased in RIG-I KO vs WT. It is surprising to have higher CD8 infiltration but no differences in tet-ova+ infiltration in Fig4 G? All the CD8 TILs are ova-tet+ which is impossible unless the MC38-ova was injected in OT-1 mice. The authors found 27% of Ova tet+ CD8 T cells in the spleen of ova bearing mice subcutaneously injected which is very unlikely. The authors should check the specificity of their tetramer staining in MC38 bearing mice or non-injected mice for example. Especially given all their CD8 T cells are ova tet+ which is impossible unless the MC38-ova was injected in OT-1 mice.

The authors show the IFNg expression by TILs without any restimulation (at least specified in the figure legend). It looks very odd as most studies use golgi inhibitors with ex vivo restimulation (PMA/I or anti-CD3) to quantify IFNg production by T cells even in tumor context.

The authors state "These data demonstrate that knocking out Rig-I in CD8+ T cells enhances their antitumor function in vivo mainly by increasing the proportion of tumor-infiltrating CD8+ T cells and the secretion of IFN- γ ." all these results don't imply a T cell intrinsic role of RIG-I and the fact that it is due to CD8 T cell increase and secretion of IFNg is not proven in this study.

Figure 5:

The link between their introduction on T cell exhaustion and the 72 hr activation with anti-CD3 is misleading. The authors should consider PD-1 has an activation marker in this context. In Figure 5A it is surprising to observe the expression of RIG-I in non-stimulated lane (0 anti-CD3) of naïve CD8 T cells. Based on Fig S1J-L, CD8 T cells from the spleen are not supposed to express RIG-I in steady state.

Figure 7:

Once again comparing full RIG-I KO with WT mice from other origin is misleading. I would suggest to analyse littermates, mixed BM chimeras, and adoptive transfer experiments to show that RIG-I expression by CD8 T cells limits anti-PD1 reactivity.

Response to Referee #1

We greatly appreciate your professional review of our manuscript. As you indicated, several issues needed to be addressed. According to your suggestions, we have extensively corrected our previous manuscript, and the detailed corrections are listed below. We first retyped your comments in *italic font* and then presented our responses to the comments to facilitate this discussion.

Comment 1:

- 102 - *In the abstract, please clarify, how RIG-I activation is induced by 'overactivated T cells'?*

Response 1:

We apologize that we did not provide a clear explanation in the Abstract of how RIG-I is upregulated by activated T cells. A recent publication in the Journal of Clinical Investigation (Jiang et al., 2023) demonstrated that after activating CD8⁺ T cells with α -CD3 and α -CD28 for 3 days, RIG-I expression is upregulated. Importantly, we observed that the upregulation of RIG-I is accompanied by the expression of PD-1. PD-1 is rapidly expressed when naïve T cells undergo TCR signalling activation. With continuous stimulation (such as in tumours or chronic infections), T cells can transition into an exhausted state, resulting in sustained high expression of PD-1. Therefore, PD-1 expression is commonly regarded as a marker of immune cell exhaustion, indicating that cells expressing high levels of PD-1 are in an exhausted or overactivated state. The stimulation time with CD3 and CD28 antibodies *in vitro* was relatively short, only three days, which we now refer to as inappropriate overactivation. For the sake of thoroughness, we have modified this description to include activated T cells (**Line 89**).

The mechanism by which activated T cells induce the upregulation of RIG-I in CD8⁺ T cells is poorly understood and is the focus of our further research. Our initial exploration focused on TCR signalling-induced T-cell activation, with detailed mechanisms requiring further investigation, possibly due to the loss of KLHL22 leading to reduced RIG-I degradation (Zhou et al., 2020) or regulation by the NFAT transcription factor family in the process of CD8⁺ T-cell exhaustion that leads to RIG-I upregulation (Martinez et al., 2015, Tillé et al., 2023). The mechanisms of RIG-I upregulation in activated CD8⁺ T cells may be diverse, and in future research, the key transcription factors and mechanisms that induce RIG-I upregulation in CD8⁺ T cells need to be investigated (**Lines 629-634**).

Comment 2:

- *The introduction has an appropriate level of detail and background.*

Response 2:

We have modified this section based on the reviewer's suggestion (**Lines 162-184**).

Comment 3:

- *Overall, the results sections are too lengthy, with much explanation and extrapolation, which should be restricted to the introduction or discussion. Introductory sentences in results should be kept as brief as necessary/possible to set the scene appropriately.*

Response 3:

We acknowledge that the Results section is overly verbose with excessive explanations or speculations. Following your advice, we have simplified the explanations and speculations in the Results section and moved them to the Introduction or Discussion (**Lines 219-222, 262-269, 319-324, 349-352, 383-394, 401-405, 413-420, 430-445, 469-473, 505-520**).

Comment 4:

• 152 - *RIG-I* is expected to be expressed in innate immune cells in addition to its observed upregulation in CD8 T cells. Did the authors examine/exclude expression in innate cells e.g. dendritic cells in tumours, which (in mice at least), also express CD8, in their analyses? The authors could look specifically at cytotoxicity using *ex vivo* CTL assays of *RIG-I* KO and WT CD8⁺ T cells.

Response 4:

The reviewer posed two questions. For the first question, the reviewer suggested that the authors conduct an *ex vivo* assay to assess the impact of *RIG-I* on the cytotoxic specificity of CD8⁺ T cells. We generated wild-type and *RIG-I* knockout OT-1 CD8⁺ T cells *ex vivo*, cocultured them with MC38-OVA tumour cells at different effector–target ratios, and observed that *RIG-I* knockout OT-1 CD8⁺ T cells exhibited

increased cytotoxicity (Figure 4A and Lines 357–362).

Figure 4. Knocking out *Rig-I* enhanced the antitumour function of CD8⁺ T cells in a transfer experiment. (A) Coculture of *Rig-I* knockout CD8⁺ T cells from OT-1 mice with MC38-OVA cells at different ratios to determine the specific killing efficiency.

Data information: Data represent different numbers (n=4 or 6) of biological replicates. The data are shown as the means \pm SEMs. Two-tailed unpaired Student's test was

used in (A). * $P < 0.05$, ** $P < 0.01$, and *** $P < 0.001$ compared with the vector or *Rig-I*^{+/+} group. Source data are available online for this figure.

We thank the reviewer for raising another interesting question about whether the authors excluded the role of *Rig-I* expression in innate immune cells. We acknowledge the role of RIG-I as an RNA sensor in innate immunity; thus, the reviewer's suggestion to explore RIG-I in innate immune cells is very reasonable. We not only explored adaptive immune cells such as CD8⁺ (CD3⁺CD8⁺) cells but also investigated innate immune cells such as NK cells and NK-like T cells. We found that *Rig-I* has a strong immunosuppressive effect on adaptive immune CD8⁺ T cells and innate immune NK-like T cells (**Appendix Figure S2A-D**) but not on NK cells (**Appendix Figure S2E-H, Lines 682-687**). Therefore, based on the above results, *Rig-I* may have a negative regulatory effect on both adaptive and innate immune cells, and we cannot rule out a specific role for *Rig-I* expression in certain types of innate immune cells. The role of tissue-resident CD8⁺ dendritic cells in antitumour immunity cannot be denied, but we focused on the impact of RIG-I on the antitumour activity of conventional CD8⁺ T cells (CD3⁺CD8⁺). We demonstrated that RIG-I directly impacts the function of conventional CD8⁺ T cells through three experiments: (1) isolating and activating naïve CD8⁺ T cells with α -CD3 and α -CD28 antibodies for 72 hours and then examining the differentiation and development of CD8⁺ T cells and the secretion of cytokines related to the antitumour function (**Figure 2D-H**); (2) coculturing *Rig-I*^{+/+} and *Rig-I*^{-/-} OT-1 CD8⁺ T cells with MC38-OVA cells for specific killing (**Figure 4A**); and (3) performing *Rig-I*^{+/+} and *Rig-I*^{-/-} CD8⁺ T-cell transfer experiments, which revealed the intrinsic regulatory role of RIG-I in CD8⁺ T cells (**Figure 4B-I**).

Appendix Figure S2. Knocking out RIG-I can enhance the functional antitumour capabilities of certain innate immune cells. (A-D) Flow cytometry analysis of CD107a and perforin levels in NKT cells from *Rig-I*^{+/+}/*Rig-I*^{-/-} mouse spleens *ex vivo*. **(E-H)** Flow cytometry analysis of CD107a and granzyme-B levels in NK cells from *Rig-I*^{+/+}/*Rig-I*^{-/-} mouse spleens *ex vivo*.

Data information: Data represent different numbers (n=6) of biological replicates. The data are shown as the means ± SEMs. Two-tailed unpaired Student's t test was used in **(B, D, E, F and H)**. A two-tailed Mann–Whitney U test was used in **(G)**. ****P* < 0.001, *****P* < 0.0001, and NS, not significant compared with the *Rig-I*^{+/+} group. Source data are available online for this figure.

Figure 2. *Rig-I* knockout enhanced the antitumour function of CD8⁺ T cells purified from the spleen *ex vivo*.

(D-H) Flow cytometry analysis of CD107a, perforin, granzyme-B, IFN- γ , and TNF- α levels in CD8⁺ T cells purified from *Rig-I*^{+/+}/*Rig-I*^{-/-} mouse spleens after α -CD3 and α -CD28 activation and stimulation with PMA/ionomycin and GolgiStop *ex vivo*.

Data information: Data represent different numbers (n=7) of biological replicates. The data are shown as the means \pm SEMs. Two-tailed unpaired Student's test was used in (D-H). **** $P < 0.0001$ compared with the *Rig-I*^{+/+} group. Source data are available online for this figure.

Figure 4. Knocking out *Rig-I* enhanced the antitumour function of CD8⁺ T cells in a transfer experiment. (A) Coculture of *Rig-I* knockout CD8⁺ T cells from OT-1 mice with MC38-OVA cells at different ratios to determine the specific killing efficiency. (B) Flowchart of the transfer of *Rig-I*^{+/+} and *Rig-I*^{-/-} mouse spleen-derived CD8⁺ T cells for the treatment of MC38 tumours. (C-E) Analysis of tumour growth curves, tumour sizes and tumour weights. (F-I) Flow cytometry analysis of the absolute number of CD8⁺ T cells infiltrating tumours, the secretion of granzyme-B and the mean fluorescence intensity after stimulation with PMA/ionomycin and GolgiStop.

Data information: Data represent different numbers (n=4 or 6) of biological replicates. The data are shown as the means ± SEMs. Two-tailed unpaired Student's test was used in (A, C, E, F, H and I). **P*<0.05, ***P*<0.01, and ****P*<0.001 compared with the vector or *Rig-I*^{+/+} group. Source data are available online for this figure.

Comment 5:

Figure 3 - The weakness of using the RIG-I KO mouse in experiments measuring tumour control, and looking specifically at CD8⁺ T cells, is that the effect of RIG-I KO on other immune cells, including innate cells is missed and the effect of RIG-I KO in CD8 T cells specifically may be over-interpreted. It is noted that differences are often subtle in these tumour models. Similarly, experiments depleting CD8 T cells

almost universally result in increased tumour progression/growth, so require careful interpretation.

Response 5:

We acknowledge the use of a mouse model with complete *Rig-I* knockout, which has the drawback of eliminating the impact of *Rig-I* on the antitumour functions of other immune cells. We adoptively transferred WT or KO *Rig-I* CD8⁺T cells and analysed their effects on tumour growth and the release of TIL-related antitumour cytokines to overcome the influence of KO mice on other immune cells. Using transfer experiments, we found that *Rig-I* knockout can partially restore the antitumour function of CD8⁺ T cells (**Figure 4**). We also observed the impact of RIG-I on the antitumour functions of innate immune cells (**Appendix Figure S2**). Hence, we have also weakened the conclusion of the specific impact of RIG-I on the antitumour function of CD8⁺ T cells (**Lines 357-377**).

Comment 6:

198 - OVA antigen can induce de novo immune responses in mice, however, this will represent only a portion of the tumour specific repertoire. Thus, the authors may be missing important information if only considering responses to this model antigen.

Response 6:

The suggestions from the reviewer are very good. The OVA antigen is only a small part of the tumour antigen library, and indeed, other antigen information is lost in the OVA tumour model. In the initial stage of our research, we used different tumour cell lines without OVA, including Hepa1-6 (**Figure 3A**), B16F10 (**Figure 3B**), and MC38 (**Figure 3C**) cells. In the added adoptive transfer experiments, we also used the MC38 tumour model (**Figure 4B-I**). These different tumour models represent the total antigen library of the corresponding tumours. As the classic research model for adaptive immune CD8⁺ T cells is based on OVA-specific antigens and the specific recognition ability of CD8⁺ T cells from OT-1 mice, we found that *Rig-I*^{-/-} CD8⁺ T

cells from *Rig-I*^{-/-} or OT-1 mice could also decrease tumour growth or specifically kill MC38-OVA tumour cells in a mouse model and specific killing assay (**Figure 3E and Figure 4A**). Therefore, our previous manuscript and the added experiments indicated that the enhancement of CD8⁺ T-cell functions through RIG-I knockout enhances the recognition of both the total tumour antigen library and the specific antigen library, as well as the killing of tumour cells.

Figure 3. *Rig-I* knockout enhanced the antitumour function of CD8⁺ T cells in animal models. (A-C) Hepa1-6, MC38 and B16F10 cells were inoculated subcutaneously into *Rig-I*^{+/+} and *Rig-I*^{-/-} mice, and the growth curves of Hepa1-6, MC38 and B16F10 tumours are shown. (E) MC38-OVA cells were inoculated subcutaneously into *Rig-I*^{+/+} and *Rig-I*^{-/-} mice, and the growth curves of MC38-OVA tumours are shown.

Data information: Data represent different numbers (n=6-9) of biological replicates. The data are shown as the means ± SEMs. Two-tailed unpaired Student's t test was used in (A). A two-tailed Mann–Whitney U test was used in (B, C, E and K). **P*<0.05, ***P*<0.01, ****P*<0.001, *****P*<0.0001, and NS, not significant compared with the *Rig-I*^{+/+} group. Source data are available online for this figure.

Comment 7:

• 222 - 'knocking out RIG-I in CD8+ T cells enhances their antitumour function in vivo mainly by increasing the proportion of CD8+ TIL and secretion of IFN γ ' - Function is not related to proportion of cells, and IFN γ is a simple marker of CD8 function, could this be quantified in a beyond one marker in flow cytometry?

Response 7:

Your suggestion is excellent, and we have modified the description in the manuscript to "Collectively, these data demonstrated that knocking out *Rig-I* in CD8⁺ T cells increased the infiltration of CD8⁺ T cells and improved their antitumour function." (Lines 374-376). In fact, IFN- γ is only one classic functional indicator of CD8⁺ T cells; in addition to IFN- γ , other markers, such as CD107a, perforin, granzyme-B and TNF- α , were used to evaluate the function of CD8⁺ T cells *ex vivo* (Figure 2D-H). In an additional adoptive transfer experiment in an animal model, we also detected another marker, granzyme-B (Figure 4G-I).

Figure 2. *Rig-I* knockout enhanced the antitumour function of CD8⁺ T cells purified from the spleen *ex vivo*. (D-H) Flow cytometry analysis of CD107a, perforin, granzyme-B, IFN- γ , and TNF- α levels in CD8⁺ T cells purified from *Rig-I*^{+/+}/*Rig-I*^{-/-} mouse spleens after α -CD3 and α -CD28 activation and stimulation with PMA/ionomycin and GolgiStop *ex vivo*.

Data information: Data represent different numbers (n=7) of biological replicates. The data are shown as the means \pm SEMs. Two-tailed unpaired Student's test was used in (D-H). **** $P < 0.0001$ compared with the *Rig-I*^{+/+} group. Source data are available online for this figure.

Figure 4. Knocking out *Rig-I* enhanced the antitumour function of CD8⁺ T cells in a transfer experiment. (G-I) Flow cytometry analysis of the absolute number of CD8⁺ T cells infiltrating tumours, the secretion of granzyme-B and the mean fluorescence intensity after stimulation with PMA/ionomycin and GolgiStop.

Data information: Data represent different numbers (n=4 or 6) of biological replicates. The data are shown as the means \pm SEMs. Two-tailed unpaired Student's test was used in (H and I). * $P < 0.05$, ** $P < 0.01$, and *** $P < 0.001$ compared with the vector or *Rig-I*^{+/+} group. Source data are available online for this figure.

Comment 8:

- 229 - "*T cell exhaustion is a type of cell population*" - rather it is a phenotype or state of *T cells*.

Response 8:

We appreciate the reviewer's suggestions and modifications regarding the details. The revised sentence is as follows: "Overactivation of T cells infiltrating the TME is a key factor leading to T-cell exhaustion, but the mechanism of CD8⁺ T-cell exhaustion caused by RIG-I remains unclear." (Lines 395-401).

Comment 9:

- 238 - Overly verbose, stick to just providing results in the results section.

Response 9:

We have simplified and changed the Results section of the manuscript (lines 219-222, 262-269, 383-394, 421-430 and 505-520).

Comment 10:

- 250 - Clarity and brevity is preferred over broad statements, particularly indicate the data the relevant data indicating the exhaustion state as opposed to looking at effector cytokine release alone.

Response 10:

Based on the reviewer's suggestion, we have edited the description of the cell exhaustion state as follows: "These data indicate that *Rig-I* can significantly enhance PD-1 expression, reduce activation marker expression and release antitumour cytokines due to activation, which attenuates the antitumour function of CD8⁺ T cells" (lines 410-413).

Comment 11:

- 256 - This is perhaps information that should be, not presented in the results section.

Based on the reviewer's suggestion, we have moved the sentences to the introduction in the Results section of the manuscript (**Lines 173-184**).

Comment 12:

- 278 - 'overactivated T cell can induce RIG-I upregulation in CD8 T cells' - this statement is confusing and should be explained, what induced RIG-I upregulation and where in the sequence of events in CD8 suppression does this occur?

Response 12:

We apologize that our description here was confusing. We have explained this statement in the manuscript, and the main content is as follows: Our data confirm that CD3 and CD28 antibodies mimic tumour antigens, stimulate TCR signalling in activated CD8⁺ T cells for 3 days, upregulate RIG-I, and are accompanied by the upregulation of PD-1. Therefore, we believe that the continuous stimulation of CD8⁺ T-cell TCR signals by tumour antigens in the TME is the leading factor causing RIG-I upregulation, and the stimulation of tumour antigens occurs at the initial stage of CD8⁺ T-cell exhaustion. An analysis of single-cell data revealed that the expression of RIG-I is mainly concentrated in the terminally exhausted CD8⁺ T cells (C13-LAG3). Therefore, we believe that the upregulation of RIG-I leads to the dysfunction of CD8⁺ T cells in the terminal stage of CD8⁺ T-cell exhaustion. (**Figure EV1G-H, Lines 237-240, Lines 464-468**).

Figure EV1. Screening single-cell sequencing data and verification of RIG-I upregulation in CD8⁺ T cells infiltrating the TME.

(G) Heatmap indicating the expression of selected gene sets in CD8⁺ T-cell subtypes infiltrating HCC and ICC. (H) Line chart showing the relative expression patterns of DDX58 in each CD8⁺ T-cell subtype.

Comment 13:

- 286 - Do they authors mean targeting RIG-I or KO of RIG-I, enhancing CD8 T cells?

Response 13:

We apologize for the mistake we have made here. We have revised this statement to "targeting of RIG-I enhances CD8⁺ T cells." (Line 478-480)

Comment 14:

- 296 - "breeding" layers - do the authors mean feeders?

Response 14:

Here, we have revised this term to "feeder cells" based on the professional suggestion of the reviewer (**Lines 481, 843, 847**).

Comment 15:

• 306 - *What is meant by" ability to weaken human tumors in vivo". Please be as specific as possible in conclusions and interpretation of results.*

Response 15:

We have provided a more detailed description of the results and explained the conclusions. Our findings demonstrate that RIG-I knockout in human CD8⁺ T cells also slows tumour growth and enhances the function of CD8⁺ T cells *in vivo* (**Lines 486-492**).

Comment 16:

• 322 - *The finding that RIG-I KO doesn't improve PD-1 efficacy in susceptible tumour models is interesting, while in resistant models it does seem to. Could the authors comment on this further regarding mechanisms of t cell suppression that differ between the two models?*

Response 16:

We also find this result very interesting. In the Discussion section, we have provided explanations and discussions on the inhibitory mechanisms of these two types of T cells as much as possible. Specifically, PD-1-sensitive tumours, which are infiltrated by many lymphocytes and upregulate PD-1, are the basis for the effectiveness of PD-1 antibodies. According to this model, the infiltration and functional activity of CD8⁺ T cells may have reached their maximum limits. This saturation state may be one of the reasons why knocking out RIG-I did not enhance the effectiveness of therapy. In contrast, for tumours insensitive to PD-1, limited lymphocyte infiltration, insignificant

PD-1 expression, and poor functional activity restrict the antitumour immune response of CD8⁺ T cells. Knocking out RIG-I can increase the infiltration of CD8⁺ T cells, enhance PD-1 expression, and strengthen their antitumour function. This finding may be a key reason why knocking out RIG-I has a significant effect on PD-1-insensitive tumours (**Lines 702-717**).

Comment 17:

1405 words for the discussion is too long in my opinion, but this may be within the journal guidelines. It could be improved with more clarity and brevity, avoiding repetition of comments and results presented already within the manuscript. At some points an unscientific writing style that is not acceptable for a peer reviewed publication is used (430). I feel the discussion needs improvement to be at the level necessary for publication.

Response 17:

Based on your overall feedback on our manuscript, we have simplified and reduced the Discussion section to more concisely and clearly address the scientific issues in this field and manuscript (**lines 582-605, 637-651, 672-676, 681-687, and 691-702**).

Comment 18:

- Some methods (505, 532, 552/3/7) need more detail, 561 requires less detail (cells were removed from the incubator). 577 - provide info on doses given and schedule.

Response 18:

Based on your suggestions, we have made appropriate additions and deletions to the Methods section (**Lines 780-783, 813, 835-851, 857-858, 871-874 and 883-890**).

Comment 19:

- 588 - animal ethics approvals MUST BE stated.

Response19:

Thank you for your suggestion. We have added the animal ethics statement (**Lines 883-890**).

Comment 20:

- Figure 1 - Why not show consistent markers across HCC and CRC tumours?

Response 20:

Thank you for your comment. In order to maintain consistency marker in HCC and CRC, we re-collected tumor tissues from five CRC patients, the secretion of granzyme-B in CD8⁺ T cells infiltrating the tumour and peritumour tissues was analysed by flow cytometry and the result was consistent with HCC (**Figure EV2H**,

Lines 251-253).

Figure EV2. The upregulation of RIG-I expression was validated in CD8⁺ T cells infiltrating the TME. (H) The expression of granzyme-B secreted by infiltrating CD8⁺ T cells in the peritumor and tumor tissues of CRC by flow cytometry after stimulating with PMA/ionomycin and GolgiStop.

Data information: Data represented different numbers (n=5) of biological replicates. Data were shown as mean \pm SEM. Two-tailed paired Student's test was used in (H).

The value of P and R was shown in the figure. *** $P < 0.001$ compared with peritumor group. Source data were available online for this figure.

Comment 21:

• Figure 4 - Axes labels should indicate the appropriate population, else how do B/C differ? Is it $CD8^+CD3^+$ and $CD8^-CD3^+$? The axes are blurry and difficult to read.

Response 21:

We apologize for any difficulty in understanding our flow cytometry data. We have relabelled the axes to present the cell populations more clearly (**Figure 3, Lines 340-346**).

Figure 3. *Rig-I* knockout enhanced the antitumour function of CD8⁺ T cells in animal models. (G-I) Statistics of the proportions of CD8⁺ T (G) and CD4⁺ T cells (H) and the ratio of CD8⁺ T/CD4⁺ T cells (I).

Data information: Data represent different numbers (n=6-9) of biological replicates. The data are shown as the means \pm SEMs. Two-tailed unpaired Student's t test was used in (A, G-J). * $P < 0.05$, ** $P < 0.01$, *** $P < 0.001$, **** $P < 0.0001$, and NS, not significant compared with the *Rig-I*^{+/+} group. Source data are available online for this figure.

Comment 22:

• Figure 6 - And in general for tumour experiments, no data on survival is provided and experiments are often not taken to a survival endpoint. How effective then is RIG-I KO at improving CD8⁺ T cell function leading to real survival benefit?

Response 22:

Thanks for your comments. In typical tumour experiments, providing specific survival benefit data would be more convincing. However, as shown in Figure 6, we aimed not only to observe the impact of RIG-I knockout on tumour growth and tumour size but also to analyse the effect of RIG-I knockout on the antitumour function of infiltrating CD8⁺ T cells. Therefore, we chose to euthanize the mice at the same time for the flow cytometry analyses of tumour size, tumour growth and infiltration of CD8⁺ T cells. Therefore, we could not obtain data on the survival benefit by euthanizing mice at the same time. However, when we subcutaneously implanted MC38 tumours in WT or KO RIG-I mice, we found that RIG-I knockout significantly prolonged the survival time of the tumour-bearing mice (**Figure EV3A-B, Lines 324-328**).

Figure EV3. *Rig-I* knockout enhanced the antitumour function. (A) Survival curves of MC38 tumour-bearing *Rig-I^{+/+}* and *Rig-I^{-/-}* mice. (B) Survival curves of MC38 tumour-bearing B-NDG mice after the adoptive transfer of *Rig-I^{+/+}* or *Rig-I^{-/-}* CD8⁺ T cells.

Data information: Data represent different numbers (n=7 for **A**, n=9 for **B**) of biological replicates. The log-rank test was used in (**A** and **B**). The data are shown as the means \pm SEMs. * $P < 0.05$, ** $P < 0.01$, and *** $P < 0.001$ compared with the *Rig-I*^{+/+} group. Source data are available online for this figure.

Comment 23:

- This is particularly relevant in the PD-1 experiments in Figure 7 in both models where survival looks achievable.

Response 23:

Thank you for your constructive suggestions. Here, we obtained survival data by reapplying PD-1 antibodies and knocking out RIG-I in CD8⁺ T cells. We found that in PD-1-sensitive tumour models, the combination of knocking out RIG-I in CD8⁺ T cells and treatment with PD-1 antibodies did not have a significant synergistic effect. However, in PD-1-insensitive tumour models, the combination of RIG-I knockout and treatment with PD-1 antibodies exerted a clear synergistic effect (**Figure 7G-H**).

Figure 7. *Rig-I* knockout combined with PD-1 monoclonal antibodies can be used to treat solid tumours that are insensitive to PD-1 antibodies. (G-H) Survival

curves of immunodeficient mice with MC38 (**G**) and B16F10 (**H**) tumours in which *Rig-I*^{+/+} or *Rig-I*^{-/-} naïve CD8⁺ T cells were transferred alone, treated with anti-PD-1 antibodies alone, or treated with their combination.

Data information: Data represent different numbers (n=10 or 11 for **G** and **H**) of biological replicates. The data are shown as the means ± SEMs. The log-rank test was used in (**G** and **H**). **P*<0.05, ***P*<0.01, ****P*<0.001, and NS, not significant compared with the other groups. Source data are available online for this figure.

References:

Jiang X, Lin J, Shangguan C, Wang X, Xiang B, Chen J, Guo H, Zhang W, Zhang J, Shi Y, Zhu J, Yang H (2023) Intrinsic RIG-I restrains STAT5 activation to modulate anti-tumor activity of CD8⁺ T cells. *The Journal of clinical investigation* 133: e160790

Zhou XA, Zhou J, Zhao L, Yu G, Zhan J, Shi C, Yuan R, Wang Y, Chen C, Zhang W, Xu D, Ye Y, Wang W, Shen Z, Wang W, Wang J (2020) KLHL22 maintains PD-1 homeostasis and prevents excessive T cell suppression. *Proceedings of the National Academy of Sciences of the United States of America* 117: 28239-28250

Martinez GJ, Pereira RM, Äijö T, Kim EY, Marangoni F, Pipkin ME, Togher S, Heissmeyer V, Zhang YC, Crotty S, Lamperti ED, Ansel KM, Mempel TR, Lähdesmäki H, Hogan PG, Rao A (2015) The transcription factor NFAT promotes exhaustion of activated CD8⁺ T cells. *Immunity* 42: 265-278

Tillé L, Cropp D, Charmoy M, Reichenbach P, Andreatta M, Wyss T, Bodley G, Crespo I, Nassiri S, Lourenco J, Leblond MM, Lopez-Rodriguez C, Speiser DE, Coukos G, Irving M, Carmona SJ, Held W, Verdeil G (2023) Activation of the transcription factor NFAT5 in the tumor microenvironment enforces CD8(+) T cell exhaustion. *Nature immunology* 24: 1645-1653

Fu Y, Urban DJ, Nani RR, Zhang YF, Li N, Fu H, Shah H, Gorka AP, Guha R, Chen L,

Hall MD, Schnermann MJ, Ho M (2019) Glypican-3-Specific Antibody Drug Conjugates Targeting Hepatocellular Carcinoma. *Hepatology (Baltimore, Md)* 70: 563-576

Zhou F, Shang W, Yu X, Tian J (2018) Glypican-3: A promising biomarker for hepatocellular carcinoma diagnosis and treatment. *Medicinal research reviews* 38: 741-767

Response to Referee #2

We appreciate your professional review of our manuscript. You identified several issues that need to be addressed. Based on your suggestions, we have thoroughly revised our previous manuscript. Below, we have rephrased your comments in *italic font* and outlined our responses to facilitate our discussion.

Comment 1:

1) The authors have fairly acknowledged the very recent work of Jiang et al. published in the JCI this year. Because of the high originality of the reported findings, the observation that similar results are obtained here is a strength of the study, and this should stimulate further research in the field. However, it remains important to test if Stat5 is involved in the context of the present work. This should be at the very least discussed.

Response 1:

Thank you for your suggestions. In the manuscript preparation stage, we noticed that very similar research has already been published in JCI. We appreciate your recognition of our work and for giving us the opportunity to discuss the differences in the mechanisms between the two research studies. RIG-I is undoubtedly a significant molecule involved in innate immunity, and many excellent studies have examined the regulation of RIG-I expression in different cells. We were particularly surprised to observe high levels of RIG-I expression in tumour-infiltrating CD8⁺ T cells and high immune checkpoint levels at the beginning of our research. Our research group, has accumulated a wealth of experience in previous studies on RIG-I, such as RIG-I inhibiting the phosphorylation of STAT3 and AKT, which regulate immune homeostasis and suppress leukaemia, respectively (Li et al., 2014, Yang et al., 2017). Furthermore, RIG-I activates the STAT1 molecule to inhibit the proliferation of leukaemia cells (Jiang et al., 2011). RIG-I can regulate cell function through multiple

molecular signals. Abnormal activation of STAT5 and AKT is closely related to tumour growth, proliferation, metastasis, and resistance, and more importantly, it is associated with the maintenance, expansion, and energy metabolism of immune cells. The work of Jiang et al. (Jiang et al., 2023) and our study confirmed that the overactivation of tumour microenvironment-infiltrating CD8⁺ T cells induces abnormal RIG-I expression, thereby inhibiting the activation of STAT5 and negatively regulating the phosphorylation of AKT at position 308, which in turn suppresses the antitumour function of CD8⁺ T cells. These two different mechanisms validate the same phenomenon, providing new insights for targeted tumour immunotherapy focusing on RIG-I molecules in CD8⁺ T cells.

By the way, Dr. Xian-Yang Li from our research group publicly discussed the hypothesis in 2019 that RIG-I might negatively regulate CD8⁺ T cell function and associated anti-tumour activity. The corresponding author of the JCI paper has acknowledged that their findings were strongly influenced by our public presentation and tested the hypothesis afterward.

Comment 2:

2) The resolution of the figures is insufficient. This is problematic to evaluate the data in Figure 1G for instance. This should be rectified.

Response 2:

Based on the reviewer's suggestion, we have increased the resolution of the figures to allow the reviewer and others to better evaluate the data (**Figure 1G**).

Figure 1. Screening single-cell sequencing data and verification of RIG-I upregulation in CD8⁺ T cells infiltrating the TME. (G) Multiple immunohistochemistry and statistical analyses of human HCC specimens; white scale bar =50 μ m.

Figure EV2. The upregulation of RIG-I expression was validated in CD8⁺ T cells infiltrating the TME. (A) Multiple immunohistochemistry and statistical analysis of human colon cancer specimens. White scale bar=50 μ m.

Comment 3:

3) These results are reminiscent of other studies published in the field demonstrating various functions for cytosolic sensors in T cells. The authors may consider discussing this aspect more, for instance the study of Li et al., Science Translational Medicine, 2020.

Response 3:

Based on the reviewer's suggestions, we discussed the article published in 2020 regarding T-cell cytoplasmic receptors. The activation of the DNA sensor STING promotes the stemness of CD8⁺ T cells and enhances their antitumour function, in contrast to the activity of the RNA sensor RIG-I, which negatively regulates the CD8⁺ T-cell antitumour function. While both RIG-I and STING can participate in regulating the function of CD8⁺ T cells in antitumour immunity, their functions are completely opposite and involve different mechanisms. The cGAS-STING pathway utilizes its own DNA-sensing function to perceive enriched genomic DNA in CD8⁺ T cells, initiating the activation of the cGAS-STING innate immune signalling pathway to prevent the excessive activation of CD8⁺ T cells and enhance their stemness and antitumour function (Li et al., 2020). On the other hand, RIG-I does not exert immunoregulatory effects on CD8⁺ T cells through its own RNA-sensing function. The study by Jiang et al. (Jiang et al., 2023) and our research confirmed that tumour antigen stimulation of the TCR in the tumour microenvironment leads to the upregulation of RIG-I, which negatively regulates the antitumour function of CD8⁺ T cells. This result represents a significant discovery in recent years in the exploration of new functions of classic innate immune molecules in adaptive immune CD8⁺ T cells, providing a new direction for future T-cell immunotherapy targeting RIG-I in combination with cGAS-STING agonists (**Lines 654-671**).

Comment 4:

4) Some typos should be corrected. For instance, in Figure 8 (dysfunctional).

In summary, this is a sound study that relies on both mouse and human data to ascribe a novel function to RIG-I. The results are relevant for cancer immunotherapy

Response 4:

Thank you for your positive feedback on our work. Based on the reviewer's suggestions, we have re-examined the manuscript and made corrections, as advised.

References:

Jiang X, Lin J, Shangguan C, Wang X, Xiang B, Chen J, Guo H, Zhang W, Zhang J, Shi Y, Zhu J, Yang H (2023) Intrinsic RIG-I restrains STAT5 activation to modulate anti-tumor activity of CD8+ T cells. *The Journal of clinical investigation* 133: e160790

Jiang LJ, Zhang NN, Ding F, Li XY, Chen L, Zhang HX, Zhang W, Chen SJ, Wang ZG, Li JM, Chen Z, Zhu J (2011) RA-inducible gene-I induction augments STAT1 activation to inhibit leukemia cell proliferation. *Proceedings of the National Academy of Sciences of the United States of America* 108: 1897-1902

Li W, Lu L, Lu J, Wang X, Yang C, Jin J, Wu L, Hong X, Li F, Cao D, Yang Y, Wu M, Su B, Cheng J, Yang X, Di W, Deng L (2020) cGAS-STING-mediated DNA sensing maintains CD8(+) T cell stemness and promotes antitumor T cell therapy. *Science translational medicine* 12: eaay9013

Li XY, Jiang LJ, Chen L, Ding ML, Guo HZ, Zhang W, Zhang HX, Ma XD, Liu XZ, Xi XD, Chen SJ, Chen Z, Zhu J (2014) RIG-I modulates Src-mediated AKT activation to restrain leukemic stemness. *Molecular cell* 53: 407-419

Yang H, Guo HZ, Li XY, Lin J, Zhang W, Zhao JM, Zhang HX, Chen SJ, Chen Z, Zhu J (2017) Viral RNA-Unprimed Rig-I Restrains Stat3 Activation in the Modulation of Regulatory T Cell/Th17 Cell Balance. *Journal of immunology (Baltimore, Md: 1950)* 199: 119-128

Response to Referee #3

We appreciate the valuable suggestions you provided on our manuscript, which have been very helpful in optimizing and enhancing its quality. According to your suggestions, we have extensively corrected our previous manuscript, and the detailed corrections are listed below. We first retyped your comments in *italic font* and then presented our responses to the comments to facilitate this discussion.

Comment 1 (on Novelty/Model System for Author):

Comparing full RIG-I KO with WT mice from other origin may be misleading with important differences in microbiome that could explain the differences in CD8 T cell differentiation and reactivity. I would suggest to analyze littermates, perform mixed BM chimeras to analyze RIG-I competent and deficient in similar environment. To address the T cell intrinsic role of RIG-I, T cell deficient using CD4cre and RIG-I^{fl/fl} or at least adoptive transfer experiments are required.

Response 1:

According to the reviewer's suggestion, analysing the antitumour capacity of RIG-I in environments similar to that of identical mice is undoubtedly a more idealized animal model, as they all have the same genetic background and microbiome, but this idealized animal model is not always achievable. Therefore, we strongly agree with your suggestion that the adoptive transfer experiment validates the intrinsic negative regulatory effect of RIG-I on T cells. We adoptively transferred WT or *Rig-I* knockout CD8⁺ T cells and analysed their effects on tumour growth and the release of TIL-related antitumour cytokines to overcome the influence of KO mice on other immune cells. We found that knocking out *Rig-I* could partially restore the antitumour function of CD8⁺ T cells via transfer experiments (**Figure 4B-I**).

Figure 4. Knocking out *Rig-I* enhanced the antitumour function of CD8⁺ T cells in a transfer experiment. (A) Coculture of *Rig-I* knockout CD8⁺ T cells from OT-1 mice with MC38-OVA cells at different ratios to determine the specific killing efficiency. (B) Flowchart of the transfer of *Rig-I*^{+/+} and *Rig-I*^{-/-} mouse spleen-derived CD8⁺ T cells for the treatment of MC38 tumours. (C-E) Analysis of tumour growth curves, tumour sizes and tumour weights. (F-I) Flow cytometry analysis of the absolute number of CD8⁺ T cells infiltrating tumours, the secretion of granzyme-B and the mean fluorescence intensity after stimulation with PMA/ionomycin and GolgiStop.

Data information: Data represent different numbers (n=4 or 6) of biological replicates. The data are shown as the means ± SEMs. Two-tailed unpaired Student's test was used in (A, C, E, F, H and I). **P*<0.05, ***P*<0.01, and ****P*<0.001 compared with the vector or *Rig-I*^{+/+} group. Source data are available online for this figure.

Comment 2:

Figure 1:

The authors should consider analyzing RIG-I expression by specific CD8 TILs subsets. Is RIG-I expressed by all TILs or only Tex subsets, which subsets T_{pex} or T_{em} Tex? The link between RIG-I expression and the low GZMB or IFN γ expression by TILs is not obvious?

Response 2:

The reviewer posed two questions. First, the authors should analyse the expression of RIG-I in the CD8 TILs subgroups to identify the main subgroups of CD8⁺ TILs expressing RIG-I. We support the reviewer's professional suggestion. Using single-cell sequencing, we analysed RIG-I expression in the CD8⁺ subgroup and found that the exhausted LAG-3 subset was the most prominent subgroup of CD8⁺ TILs expressing RIG-I (Figure EV1G-H).

Figure EV1. Screening single-cell sequencing data and verification of RIG-I upregulation in CD8⁺ T cells infiltrating the TME. (G) Heatmap indicating the expression of selected gene sets in CD8⁺ T-cell subtypes infiltrating HCC and ICC. **(H)** Line chart showing the relative expression patterns of DDX58 in each CD8⁺ T-cell subtype.

Second, the authors should analyse the relationship between RIG-I expression and low expression of GZMK and IFN- γ in CD8⁺ TILs. We reanalysed the relationship between RIG-I expression in CD8⁺ TILs and the expression of GZMB or IFN- γ in tumour single-cell sequencing data and found no correlation between RIG-I expression and GZMK or IFN- γ expression in CD8⁺ TILs (**Figure EV2I-J**).

Figure EV2. The upregulation of RIG-I expression was validated in CD8⁺ T cells infiltrating the TME. (G) The correlation analysis of DDX58 expression with GZMB and IFNG expression.

Data information: Pearson correlation test was used in (G). The value of P and R was shown in the figure.

Comment 3:

Figure 2:

The RNAseq analysis is interesting but not very well detailed in the manuscript. Before focusing on key genes and pathways, the authors should first show the raw differential gene expression between RIG Wt and Ko (volcano plot showing the adj p value and FC) for example. Why did the authors select these genes in particular in Fig 2B? Are they handpicked, higher FC or p value or related to a pathway enriched in Fig2A?

It is not clear which markers and gates were used to quantify the memory and effector cells in Fig 2K-L or in Figure S3. Especially, with regards to the CD44-CD62L- pop present in the representative FACS plots in Fig2I?

The authors found a decrease in naïve in FigS3A but NS in Fig2J.

The authors state "To rule out the possibility that Rig-I indirectly regulates the antitumor function of CD8+ T cells through other immune cells, we purified CD8+ T cells from Rig-I+/+ and Rig-I-/- mouse spleens in vitro (Fig. S4). » While it rules out a bystander cell role during the PMA/ionomycin stimulation, it doesn't prove a T cell intrinsic role of RIG-I in CD8 T lymphocytes in the regulation of T cells.

The authors should prove that RIG-I is expressed by T cells in the spleen of WT mice. In Fig S1 they show RIG one is absent which suggest that the differences observed in the spleen are due to RIG-I expression by other cells. In FIG1 and S1 the authors suggest RIG-I is specific for TILs. Thus, the difference in cytokine production upon PMA/ionomycin stimulation might just reflect the higher memory frequency in RIG-I-/- mice.

For the Fig S4 the authors state"Analysis of the IFN- γ , granzyme-B and the development (including naive, memory and effector cells) of CD8+ T or CD8-T cells purified from Rig-I+/+/Rig-I-/-spleen after stimulation with PMA, ionomycin and Golgi blockers in vitro by flow cytometry." but the figure doesn't show purified cells and no analysis of CD8- cells???. This figure looks similar to the figure S2 in terms of settings and gating.

Comparing full RIG-I KO with WT mice from other origin may be misleading with important differences in microbiome that could explain the differences in CD8 T cell differentiation and reactivity. I would suggest to analyse littermates or mixed BM chimeras to analyse RIG-I competent and deficient in similar environment.

Response 3:

Our responses to the seven questions raised by the reviewer are listed below.

(1) The first question suggests that before focusing on differentially expressed genes and signalling pathways, the authors should thoroughly present all differential genes between the WT and KO groups. Why were the specific differentially expressed genes chosen in Figure 2B? Were they manually selected, or did they have higher FC or P values? Or were they enriched in the signalling pathways shown in Fig2A? We aimed to present the bioinformatics analysis process as comprehensively as possible by displaying all differentially expressed genes and signalling pathways between the WT and KO groups to address these concerns, as shown in the figure (**Appendix Figure S1**) and dataset (**Dataset EV1**). On the left side of **Figure 2A**, we selected signalling pathways significantly different from those in **Dataset EV1**. The differentially

expressed genes shown in **Figure 2A on the right** are from signalling pathways associated with cytotoxicity, cell killing, and interferon production, as shown in **Figure 2A on the left**. We have also added a description of this analysis in the manuscript and figure legend (**Lines 268-277, and 1124-1130**).

Appendix Figure S1. Volcano plot showing differentially expressed genes (DEGs) in *Ddx58*^{-/-} vs. *Ddx58*^{+/+} models. The upregulated genes (red) and the downregulated genes (blue) with a fold change ≥ 2 and with $P_{\text{adjust}} < 0.05$ are shown.

(2) The second question suggests that the authors clearly label the memory and effector cells. We apologize for not clearly labelling the memory and effector cells in Figure 2. We have now prominently labelled these cell populations in the corresponding figures to better show our data (**Figure 2B**).

(3) The third question indicates that in Fig. S3A, the naïve population decreased, while in Fig. 2J, the proportion was not statistically significant. We increased the sample size, repeated the experiment, and found that the proportion of naïve cells was still decreased (**Figure 2C**).

(4) The fourth question indicated that the authors did not demonstrate the intrinsic role of *Rig-I* in regulating CD8⁺ T lymphocytes. We acknowledge that the use of PMA/ionomycin stimulation does not prove the intrinsic role of *Rig-I* in CD8⁺ T cells. We isolated naïve CD8⁺ T cells, activated them with α -CD3 and α -CD28 antibodies

for 72 hours, and then examined the differentiation (**Figure 2B-C**) and secretion of cytokines with antitumour functions (**Figure 2D-H**) from CD8⁺ T cells to confirm the intrinsic role of RIG-I in regulating CD8⁺ T cells.

Figure 2. *Rig-I* knockout enhanced the antitumour function of CD8⁺ T cells purified from the spleen *ex vivo*. (A) RNA-seq and KEGG enrichment analysis of signalling pathways (left panel) and key genes associated with leukocyte-mediated cytotoxicity, regulation of cell killing and interferon-gamma production (right panel) in CD8⁺ T cells from *Rig-I*^{+/+}/*Rig-I*^{-/-} mouse spleens. (B-C) Flow cytometry analysis of the development (including naïve, central memory and effector cells) of CD8⁺ T

cells purified from *Rig-I*^{+/+}/*Rig-I*^{-/-} mouse spleens after α -CD3 and α -CD28 activation *ex vivo*. **(D-H)** Flow cytometry analysis of CD107a, perforin, granzyme-B, IFN- γ , and TNF- α levels in CD8⁺ T cells purified from *Rig-I*^{+/+}/*Rig-I*^{-/-} mouse spleens after α -CD3 and α -CD28 activation and stimulation with PMA/ionomycin and GolgiStop *ex vivo*.

Data information: Data represent different numbers (n=7) of biological replicates. The data are shown as the means \pm SEMs. Two-tailed unpaired Student's test was used in **(C-H)**. **** $P < 0.0001$ compared with the *Rig-I*^{+/+} group. Source data are available online for this figure.

(5) In the fifth question, the reviewer suggested that the authors should demonstrate the expression of *Rig-I* in T cells from WT mouse spleens. Figure S1 shows that RIG-I is absent, indicating that the observed differences in the spleen are caused by other cells expressing RIG-I. Figures 1 and S1 show the specificity of RIG-I for TILs. In fact, we confirmed *Rig-I* expression in naïve **(Figure 5A)** or total **(Figure 5F)** CD8⁺ T cells in the spleen of WT mice. Additionally, RIG-I was also expressed in total CD8⁺ T cells among human PBMC CD8⁺ T cells **(Figure 5G-H)**. Furthermore, we believe that the reviewer misunderstood our data. Due to the increased expression of *Rig-I* in TILs, our Western blot analysis did not reveal excessive expression, making the expression of RIG-I in the spleen appear weak (previously shown in Figure S1J) to the point of not being visible. We repeated the experiment and found that CD8⁺ T cells in tumours express *Rig-I* at significantly higher levels than those in the spleen **(Figure EV2D)**. Therefore, the expression of RIG-I is not tissue-specific but is upregulated in TILs, especially in CD8⁺ T cells.

Figure 5. The activation of CD8⁺ T cells induced the upregulation of *Rig-I*, inhibiting the PI3K/AKT/glycolysis signalling pathway to counteract its antitumour function. (A) Naïve CD8⁺ T cells were isolated from the spleens of wild-type mice, and after 72 hours of costimulation with α -CD3/ α -CD28, *Rig-I* expression was detected using Western blotting. (F) After 72 hours of costimulation with α -CD3/ α -CD28, the expression of RIG-I, p-AKT (Thr308), AKT, and GAPDH in WT or KO *Rig-I* CD8⁺ T cells from mouse spleens was detected using Western blotting. (G) Validation of the efficiency of RIG-I knockout in human CD8⁺ T cells. (H) After knocking out RIG-I, the expression of RIG-I, p-AKT (Thr308), AKT, and GAPDH in human CD8⁺ T cells was detected using Western blotting.

Figure EV2. The upregulation of RIG-I expression was validated in CD8⁺ T cells infiltrating the TME. (D-F) The expression of *Rig-I* in CD8⁺ T cells from the spleens and tumours of different tumour-bearing mice was detected using Western blotting.

(6) In the sixth question, the reviewer inquired about why the previous Figure S4 did not display purified cells and about the absence of an analysis of CD8⁺ T cells. We are very grateful for the reviewer's careful identification of our mistake in the description. We apologize for the error in our description. Here, we analysed only IFN- γ levels,

granzyme-B levels, and the development (including naïve, central memory, and effector cells) of purified CD8⁺ T cells (**previous Figure S4**). In response to the reviewer's suggestion to investigate the intrinsic role of RIG-I in CD8⁺ T cells, we improved our methodology and description by activating CD8⁺ T cells with α -CD3 and α -CD28 antibodies for 72 hours to assess the differentiation, development, and secretion of cytokines with antitumour functions (**Figure 2B-H, Lines 292-298**). Consequently, we have replaced the data in the **previous Figure S4**.

(7) In the seventh question, the reviewer suggested analysing RIG-I capabilities and defects in littermates or mixed bone marrow chimaeras to assess them in a similar environment. We agree with the reviewer's suggestion that littermates or mixed bone marrow chimaeras are ideal animal models, but they may not always be feasible for practical experiments. Based on the reviewer's suggestion, we conducted transfer experiments to confirm the regulatory role of RIG-I in the antitumour function of CD8⁺ T cells (**Figure 4B-I**).

Comment 4:

Figure 3:

The authors show in well characterized CD8 T cells dependent models that RIG-I -/- mice have lower tumor growth and conclude "Collectively, these findings underscore the direct impact of Rig-I in limiting the antitumor efficacy of CD8+ T cells." This is an overstatement given the authors already showed developmental differences in steady state conditions in RIG-I KO vs WT (Fig2). In addition, the CD8 T cell dependence of the model doesn't imply an intrinsic effect of RIG-I in CD8 T cells.

Response 4:

We acknowledge that the conclusions shown in Figure 3 have been overstated; therefore, we have weakened our conclusion that *Rig-I* is involved in the negative regulation of the antitumour function of CD8⁺ T cells (**Lines 338-340**). Moreover, the

CD8⁺ T-cell dependency in this model does not imply an intrinsic impact of RIG-I on CD8⁺ T cells. We (1) isolated and activated naïve CD8⁺ T cells with α -CD3 and α -CD28 antibodies for 72 hours and then examined the differentiation and development of CD8⁺ T cells and the secretion of cytokines with antitumour functions; (2) cocultured *Rig-I*^{+/+} and *Rig-I*^{-/-} OT-1 CD8⁺ T cells with MC38-OVA cells for specific killing; and (3) performed *Rig-I*^{+/+} and *Rig-I*^{-/-} CD8⁺ T-cell transfer experiments, which demonstrated the intrinsic regulatory effect of RIG-I on CD8⁺ T cells, to address this issue (**Figure 2B-I, Figure 4A-I**).

Comment 5:

Figure 4:

The authors should consider showing cell numbers in addition to cell frequencies. They show no differences in CD8+% in figure 4A (31,9 vs 34,2%) and yet in fig 4B CD8+ T cells are shown increased in RIG-I KO vs WT.

It is surprising to have higher CD8 infiltration but no differences in tet-ova+ infiltration in Fig4 G? All the CD8 TILs are ova-tet+ which is impossible unless the MC38-ova was injected in OT-1 mice. The authors found 27% of Ova tet+ CD8 T cells in the spleen of ova bearing mice subcutaneously injected which is very unlikely. The authors should check the specificity of their tetramer staining in MC38 bearing mice or non-injected mice for example. Especially given all their CD8 T cells are ova tet+ which is impossible unless the MC38-ova was injected in OT-1 mice.

The authors show the IFN γ expression by TILs without any restimulation (at least specified in the figure legend). It looks very odd as most studies use golgi inhibitors with ex vivo restimulation (PMA/I or anti-CD3) to quantify IFN γ production by T cells even in tumor context.

The authors state "These data demonstrate that knocking out Rig-I in CD8+ T cells enhances their antitumor function in vivo mainly by increasing the proportion of tumor-infiltrating CD8+ T cells and the secretion of IFN- γ ." all these results don't

imply a T cell intrinsic role of RIG-I and the fact that it is due to CD8 T cell increase and secretion of IFN γ is not proven in this study.

Response 5:

There are several issues to address.

(1) First, the reviewer suggested that in addition to focusing on cell proportions, the authors should also consider cell numbers. In our adoptive transfer experiment, we used counting beads to quantify the number of CD8⁺ T cells infiltrating tumours in *Rig-I* knockout mice, revealing that *Rig-I* knockout increased their infiltration within

tumours (**Figure 4F**).

Figure 4. Knocking out *Rig-I* enhanced the antitumour function of CD8⁺ T cells in a transfer experiment. (F) Flow cytometry analysis of the absolute number of CD8⁺ T cells infiltrating tumours, the secretion of granzyme-B and the mean fluorescence intensity after stimulation with PMA/ionomycin and GolgiStop.

Data information: Data represent different numbers (n=4 or 6) of biological replicates. The data are shown as the means \pm SEMs. Two-tailed unpaired Student's test was used in (F). * $P<0.05$, ** $P<0.01$, and *** $P<0.001$ compared with the vector or *Rig-I*^{+/+} group. Source data are available online for this figure.

(2) Second, we selected a new representative image showing differences in CD8⁺ T-cell percentages (19.6% vs. 34.2%) (**Figure 3F**).

Figure 3. *Rig-I* knockout enhanced the antitumour function of CD8⁺ T cells in animal models. (F) Flow cytometry analysis of the IFN- γ level and proportion of CD8⁺ T/CD4⁺ T cells infiltrating tumours formed from the colon cancer cell line MC38 in *Rig-I*^{+/+} and *Rig-I*^{-/-} mice after stimulation with PMA/ionomycin and GolgiStop.

(3) Third, we have reconsidered the logic of Fig. 4G and suspect that the data may be inaccurate due to nonspecific staining with the OVA antibody. Therefore, for rigor, we have removed the previous data from the **previous version of Figure 4G**. We conducted (1) *Rig-I*^{+/+} and *Rig-I*^{-/-} OT-1 CD8⁺ T-cell cocultures with MC38-OVA cells for specific killing and (2) *Rig-I*^{+/+} and *Rig-I*^{-/-} CD8⁺ T-cell transfer experiments, which demonstrated that knocking out *Rig-I* in CD8⁺ T cells increased the infiltration of CD8⁺ T cells and improved their antitumour function, to support this conclusion (**Figure 4A, F-I**).

Figure 4. Knocking out *Rig-I* enhanced the antitumour function of CD8⁺ T cells in a transfer experiment. (A) Coculture of *Rig-I* knockout CD8⁺ T cells from OT-1

mice with MC38-OVA cells at different ratios to determine the specific killing efficiency. **(F-I)** Flow cytometry analysis of the absolute number of CD8⁺ T cells infiltrating tumours, the secretion of granzyme-B and the mean fluorescence intensity after stimulation with PMA/ionomycin and GolgiStop.

Data information: Data represent different numbers (n=4 or 6) of biological replicates. The data are shown as the means \pm SEMs. Two-tailed unpaired Student's test was used in **(A, F, H and I)**. * $P < 0.05$, ** $P < 0.01$, and *** $P < 0.001$ compared with the vector or *Rig-I*^{+/+} group. Source data are available online for this figure.

(4) Fourth, the reviewer noted that the authors did not stimulate the release of cytokines from tumour-infiltrating CD8 cells in the figure legend. We apologize for not clearly describing in the figure legend that all cytokine detection assays in the manuscript were performed using cells stimulated with PMA/ionomycin and GolgiStop. We have amended the description in the figure legends and Methods **(Lines 1098, 1132, 1166, 1169, 1206, 1242, 1264-1265, 1348, 1368 and 1390)**.

(5) Fifth, the reviewer noted that the intrinsic role of *Rig-I* in CD8⁺ T cells was not confirmed. We (1) isolated and activated naïve CD8⁺ T cells with α -CD3 and α -CD28 antibodies for 72 hours and then examined the differentiation and development of CD8⁺ T cells and the secretion of cytokines with antitumour functions; (2) cocultured *Rig-I*^{+/+} and *Rig-I*^{-/-} OT-1 CD8⁺ T cells with MC38-OVA cells for specific killing; and (3) performed *Rig-I*^{+/+} and *Rig-I*^{-/-} CD8⁺ T-cell transfer experiments, which demonstrated the intrinsic regulatory effect of RIG-I on CD8⁺ T cells, to address this issue **(Figure 2B-I, Figure 4A-I)**.

Comment 6:

Figure 5:

The link between their introduction on T cell exhaustion and the 72 hr activation with anti-CD3 is misleading. The authors should consider PD-1 has an activation marker

in this context. In Figure 5A it is surprising to observe the expression of RIG-I in non-stimulated lane (0 anti-CD3) of naïve CD8 T cells. Based on Fig S1J-L, CD8 T cells from the spleen are not supposed to express RIG-I in steady state.

Response 6:

There are two issues.

(1) First, we acknowledge that in our 72-hour activation model, referring to PD-1 expression as exhaustion in T cells is inaccurate. In this short-term activation model, PD-1 expression should be referred to as an activation marker. We have changed the description in the manuscript (**Lines 380, 400, 553, 614, 619, 630, and 662**).

(2) Second, due to the presentation of data in the **previous Figure S1J-L**, which caused a misunderstanding by the reviewer, importantly, CD8⁺ T cells in the spleen of wild-type mice express *Rig-I*, as shown in **Figure 5A and F** and in a previously published JCI article (Jiang, Lin et al., 2023). However, the expression of *Rig-I* is upregulated in tumour-infiltrating CD8⁺ T cells. Due to limitations in the Western blot analysis, the expression of RIG-I in CD8⁺ T cells in the spleen may appear low or absent. Therefore, we repeated the experiment described in the original **Figure S1J** with an increased exposure time and observed an increase in *Rig-I* expression in infiltrating CD8⁺ T cells (**Figure EV2D**).

Figure 5. The activation of CD8⁺ T cells induced the upregulation of *Rig-I*, inhibiting the PI3K/AKT/glycolysis signalling pathway to counteract its

antitumour function. (A) Naïve CD8⁺ T cells were isolated from the spleens of wild-type mice, and after 72 hours of costimulation with α -CD3/ α -CD28, *Rig-I* expression was detected using Western blotting (F) After 72 hours of costimulation with α -CD3/ α -CD28, the expression of RIG-I, p-AKT (Thr308), AKT, and GAPDH in WT or KO *Rig-I* CD8⁺ T cells from mouse spleens was detected using Western blotting.

Figure EV2. The upregulation of RIG-I expression was validated in CD8⁺ T cells infiltrating the TME. (D-F) The expression of *Rig-I* in CD8⁺ T cells from the spleens and tumours of different tumour-bearing mice was detected using Western blotting. **Comment 7:**

Figure 7:

Once again comparing full RIG-I KO with WT mice from other origin is misleading. I would suggest to analyse littermates, mixed BM chimeras, and adoptive transfer experiments to show that RIG-I expression by CD8 T cells limits anti-PD1 reactivity.

Response 7:

We appreciate the reviewer's constructive feedback on the combination therapy model. We acknowledge the limitations of this knockout model. Here, we obtained survival data by reapplying PD-1 antibodies and knocking out RIG-I in CD8⁺ T cells. We found that in PD-1-sensitive tumour models, the combination of knocking out RIG-I in CD8⁺ T cells and administering PD-1 antibodies did not have a significant synergistic effect. However, in PD-1-insensitive tumour models, the combination of knocking out RIG-I and administering PD-1 antibodies exerted a clear synergistic effect (**Figure 7G-H**).

Figure 7. *Rig-I* knockout combined with PD-1 monoclonal antibodies can be used to treat solid tumours that are insensitive to PD-1 antibodies. (G-H) Survival curves of immunodeficient mice with MC38 (G) and B16F10 (H) tumours in which *Rig-I*^{+/+} or *Rig-I*^{-/-} naïve CD8⁺ T cells were transferred alone, treated with anti-PD-1 antibodies alone, or treated with their combination.

Data information: Data represent different numbers (n=10 or 11 for G and H) of biological replicates. The data are shown as the means \pm SEMs. The log-rank test was used in (G and H). * P <0.05, ** P <0.01, *** P <0.001, and NS, not significant compared with the other groups. Source data are available online for this figure.

References:

Jiang X, Lin J, Shangguan C, Wang X, Xiang B, Chen J, Guo H, Zhang W, Zhang J, Shi Y, Zhu J, Yang H (2023) Intrinsic RIG-I restrains STAT5 activation to modulate anti-tumor activity of CD8⁺ T cells. *The Journal of clinical investigation* 133: e160790

31st Jul 2024

Dear Prof. Lu,

Thank you for the submission of your revised manuscript to EMBO Molecular Medicine. We have now received the enclosed reports from the referees that were asked to re-assess it. As you will see the reviewers are now globally supportive and I am pleased to inform you that we will be able to accept your manuscript pending the following final amendments:

- 1) Please note that all corresponding authors are required to supply an ORCID ID for their name. Currently ORCID IDs are missing for Duan, Sun, and Li.
 - 2) Please include the specific links for the datasets in the Data availability section (i.e. for GEO GSE230045 dataset and GSA dataset CRA010683). We suggest authors to format the Data availability statement according to the example below:
"The datasets and computer code produced in this study are available in the following databases:
- Chip-Seq data: Gene Expression Omnibus GSE46748 (<https://www.ncbi.nlm.nih.gov/geo/query/acc.cgi?acc=GSE46748>)
- Modeling computer scripts: GitHub (<https://github.com/SysBioChalmers/GECKO/releases/tag/v1.0>)
- [data type]: [full name of the resource] [accession number/identifier] ([doi or URL or identifiers.org/DATABASE:ACCESSION])"
 - 3) Data Availability: Please move the Data Availability section to the end of the Methods section
 - 4) Please rename "Conflict of interest statement" to "Disclosure and competing interests statement". We updated our journal's competing interests policy in January 2022 and request authors to consider both actual and perceived competing interests. Please review the policy <https://www.embopress.org/competing-interests> and update your competing interests if necessary.
 - 5) References: Please correct the reference citation in the reference list. Where there are more than 10 authors on a paper, note that only 10 should be listed, followed by "et al.". Currently there are more than 10 authors listed before et al. Please check "Author Guidelines" for more information.
<https://www.embopress.org/page/journal/17574684/authorguide#referencesformat>
 - 6) In the Methods, please take care of the following:
 - Animals: We would request additional information on the animals and their housing conditions - specifically information such as light/dark cycles and which experiments were associated with which specific gender.
 - Cell lines: Please include all information requested in the author checklist for cell lines used in the manuscript (accession number in repository or supplier name, catalog number, clone number, and/or RRID). Please also be sure to include a sentence in the Methods as to whether or not the cell lines were recently authenticated.
 - Please ensure that a statement on whether or not blinding was done is included in the Methods even if no blinding was done.
 - 7) All materials and methods need to be described in the main text using our 'Structured Methods' format, which is required for all research articles. According to this format, the Methods section includes a Reagents and Tools Table (listing key reagents, experimental models, software and relevant equipment and including their sources and relevant identifiers) followed by a Methods and Protocols section describing the methods using a step-by-step protocol format. The aim is to facilitate adoption of the methodologies across labs. More information on how to adhere to this format as well as a downloadable template (.docx) for the Reagents and Tools Table can be found in our author guidelines:
<https://www.embopress.org/page/journal/17574684/authorguide#structuredmethods>
- An example of a Method paper with Structured Methods can be found here:
<https://www.embopress.org/doi/10.15252/msb.20178071>.
- 8) Please place individual sections of the manuscript in the following order: Title page - Abstract & Keywords - Introduction - Results - Discussion - Methods - Data Availability - Acknowledgements - Disclosure and Competing Interests Statement - The Paper Explained - References - Figure Legends - Expanded View Figure Legends.
 - 9) For the figures and figure legends, please take care of the following:
 - Please note that the legend for figure 5i-l is mislabeled as 5i-k in the manuscript. This needs to be rectified.
 - Please note that the exact p values are not provided in the legends of figures 1h-l; 2c-h; 3a, g; 5i-l; 6b; EV 4b, f.
 - Please indicate the statistical test used for data analysis in the legends of figures 2a.
 - Please note that in figures 1j-l; EV 3d-e; there is a mismatch between the annotated p values in the figure legend and the annotated p values in the figure file that should be corrected.
 - Please note that the scale bar is missing for figure 1g.
 - Please add a heading for "Expanded View Figure Legends" after the main figure legends.
 - 10) Dataset: Dataset EV1 should have a legend within the file in a separate tab (not in the main manuscript file).
 - 11) Funding: Please note that funding information should be given in the "Acknowledgements" section (not in its own separate section). Please ensure that all funding sources are entered into the manuscript submission system - currently this information is incomplete.
 - 12) Source Data: Please ensure that the Source Data are uploaded as a single source data file (zipped) per figure, with the panels clearly visible in the folder structure. In addition, it would be helpful if the excel files for Figures 1, 4, and EV3 were renamed, as currently it is not possible to open them on our computers without renaming the files.
 - 13) Source Data: Please also double check the Source Data for Figure 5, right panel of L. The numbers in the Source Data file do not seem to match the figure. If you change the figure, please briefly inform us what changes were made and the reason, and

whether any conclusions are changed.

14) The Paper Explained: Please do not upload "The Paper Explained" separately and rather include it in the main manuscript text just after the Disclosure and Competing Interests Statement.

15) As part of the EMBO Publications transparent editorial process initiative (see our policy here:

https://www.embopress.org/transparent-process#Review_Process), EMBO Molecular Medicine will publish online a Peer Review File (PRF) to accompany accepted manuscripts. This file will be published in conjunction with your paper and will include the anonymous referee reports, your point-by-point response and all pertinent correspondence relating to the manuscript. Let us know whether you agree with the publication of the PRF and as here, if you want to remove or not any figures from it prior to publication. Please note that the Authors checklist will be published at the end of the PRF.

16) Please provide a point-by-point letter INCLUDING my comments as well as the reviewer's reports and your detailed responses (as Word file).

I look forward to reading a new revised version of your manuscript as soon as possible

Yours sincerely,

Poonam Bheda

Poonam Bheda, PhD
Scientific Editor
EMBO Molecular Medicine

***** Reviewer's comments *****

Referee #1 (Comments on Novelty/Model System for Author):

This manuscript focuses on demonstrating a new role for the retinoic acid-inducible I (RIG-I) in the inhibition of CD8+ T cell function. The authors hypothesize that RIG-I functions as an intracellular checkpoint that limits anti-tumour CD8+ T cell immunity. They show that RIG-I expression is enhanced in CD8 TILs in single cell datasets from different human cancers. They use a RIG-I KO mouse model to assess the effects of RIG-I on splenic CD8 T cells phenotype using RNA sequencing and flow cytometry. Using these same mice, the authors show that tumours develop slower when RIG-I is not present (KO vs. WT). They show that the CD8 T cells in tumour-bearing RIG-I KO mice are increased in number and production of effector cytokines, including antigen-specific CD8 T cells in the tumour. They show by western blot that RIG-I KO enhances pAKT and enhances CD8 T cell activation, which is targetable with small molecule inhibitors of RIG-I. Overall, the data presented in the manuscript is novel, interesting to the field and well produced with appropriate controls and experimental rigor to answer experimental questions.

The authors have taken time to conduct several extended analyses and experiments at the request of the reviewers, and modified the text to improve the manuscript. While I agree with reviewer 3 that the original model (full RIG-I KO vs WT) was not ideal and a floxed mouse would have been better to investigate KO of RIG-I in CD8 T cells specifically if the authors have limited their focus solely to analysis of CD8 T cell function in this setting I believe the data may still be of interest and further detailed analysis could be the topic of further publications. Given the minimal impact on overall survival the utility of targeting RIG-I in combination with standard of care immunotherapy remains to be determined.

Referee #1 (Remarks for Author):

The authors have taken the time to complete addition work requested and improved the manuscript for which I thank them.

Referee #2 (Remarks for Author):

The authors have addressed my concerns.

Referee #3 (Remarks for Author):

The authors thoroughly addressed all the points raised by the reviewers and made important improvement in their revised manuscript to uncover the negative role of RIG-1 in CD8 T cell anti-tumor functions.

The authors addressed the minor editorial issues.

29th Aug 2024

Dear Prof. Lu,

Congratulations on an excellent manuscript, I am pleased to inform you that your manuscript has been accepted for publication in the EMBO Molecular Medicine. Thank you for your comprehensive response to referee concerns and for providing detailed source data. It has been a pleasure to work with you to get this to the acceptance stage.

Yours sincerely,

Poonam Bheda, PhD
Scientific Editor
EMBO Molecular Medicine
